# The temperature dependence of ice-nucleating particle concentrations affects the radiative properties of tropical convective cloud systems

**Rachel E. Hawker*[1], Annette K. Miltenberger[1, a], Jonathan M. Wilkinson[2], Adrian A. Hill[2], Ben J. Shipway[2], Zhiqiang Cui[1], Richard J. Cotton[2], Ken S. Carslaw[1], Paul R. Field[1, 2], Benjamin J. Murray[1].**

1. Institute for Climate and Atmospheric Science, University of Leeds, Leeds, LS2 9JT, United Kingdom.
2. Met Office, Exeter, EX1 3PB, United Kingdom.
a. now at : Institute for Atmospheric Physics, Johannes Gutenberg University Mainz, Mainz, 55128, Germany.

*Correspondence to:* Rachel E. Hawker (eereh@leeds.ac.uk)

## Abstract

Convective cloud systems in the maritime tropics play a critical role in global climate, but accurately representing aerosol interactions within these clouds persists as a major challenge for weather and climate modelling. We quantify the effect of ice-nucleating particles (INP) on the radiative properties of a complex Tropical Atlantic deep convective cloud field using a regional model with an advanced double-moment microphysics scheme. Our results show that the domain-mean daylight outgoing radiation varies by up to 18 W m$^{-2}$ depending on the chosen INP parameterisation. The key distinction between different INP parameterisations is the temperature dependence of ice formation, which alters the vertical distribution of cloud microphysical processes. The controlling effect of the INP temperature dependence is substantial even in the presence of Hallett-Mossop secondary ice production, and the effects of secondary ice formation depend strongly on the chosen INP parameterisation. Our results have implications for climate model simulations of tropical clouds and radiation, which currently do not consider a link between INP particle type and ice water content. The results also provide a challenge to the INP measurement community, since we demonstrate that INP concentration measurements are required over the full mixed-phase temperature regime, which covers around 10 orders of magnitude.

# 1. Introduction

Deep convective clouds are important drivers of local, regional and global climate and weather (Arakawa, 2004; Lohmann et al., 2016). They produce substantial precipitation (Arakawa, 2004) and the associated phase changes release latent heat that helps to drive global atmospheric circulation (Fan et al., 2012). Convective clouds have a direct impact on climate through interactions with incoming shortwave and outgoing longwave radiation (Lohmann et al., 2016), for example by producing radiatively important long-lived cirrus clouds (Luo and Rossow, 2004). The clouds extend from the warmer lower levels of the atmosphere where only liquid exists to the top of the troposphere where only ice exists (Lohmann et al., 2016). Between these levels is the mixed-phase region where both liquid and ice coexist and interact (Seinfeld and Spyros, 2006). Within the mixed-phase region, primary ice particles can form heterogeneously through the freezing of cloud droplets by ice-nucleating particles (INP). The importance and relative contribution of heterogeneous freezing to ice crystal number concentrations (ICNC) and resultant cloud properties, such as cloud reflectivity, is very uncertain (Cantrell and Heymsfield, 2005; Kanji et al., 2017). This uncertainty stems from the difficulty of predicting INP number concentrations (Kanji et al., 2017; Lacher et al., 2018) as well as the difficulty of quantifying complex interactions between heterogeneous freezing and other ice production mechanisms (Crawford et al., 2012; Huang et al., 2017; Phillips et al., 2005).

Understanding the effects of INP on convective clouds presents substantial challenges. Measurements indicate that INP number concentrations can vary by as much as six orders of magnitude at any one temperature due to variations in, for example, aerosol source, chemical or biological composition, and surface morphology (DeMott et al., 2010; Kanji et al., 2017). Large variability exists even in measurements of individual regions or aerosol populations (Boose et al., 2016b; Kanji et al., 2017; Lacher et al., 2018). For example, there are four orders of magnitude variation in summertime measurements of INP number concentrations in the Saharan Air Layer at -33°C (Boose et al., 2016b). Even for particles of similar and known mineralogy, measurements of ice-nucleation efficiency can span several orders of magnitude: The spread in laboratory measurements of ice nucleation active site densities ($n_s$) for different types of feldspar spans seven orders of magnitude at -15°C (Atkinson et al., 2013; Harrison et al., 2016, 2019; Peckhaus et al., 2016). Our ability to understand and quantify such variability in INP concentrations has improved as more measurements have been made. Although INP concentrations do not simply correlate with meteorological variables such as pressure and temperature (Boose et al., 2016a; Lacher et al., 2018; Price et al., 2018), aerosol surface area (Lacher et al., 2018) and diameter (DeMott et al., 2015) provide some predictability and global models based on

known INP-active materials show reasonable skill in simulating global INP concentrations (Shi and Liu, 2019; Vergara-Temprado et al., 2017).

It is known from model simulations that changes in INP number concentration affect the microphysical properties and behaviour of deep convective clouds (Deng et al., 2018; Fan et al., 2010a, 2010b; Gibbons et al., 2018; Takeishi and Storelvmo, 2018). However, in these model studies perturbations to INP number concentrations have predominantly involved uniform increases in aerosol or INP concentrations with all simulations using the same INP parameterisation (Carrió et al., 2007; Connolly et al., 2006; Deng et al., 2018; Ekman et al., 2007; Fan et al., 2010a; Gibbons et al., 2018; van den Heever et al., 2006; Phillips et al., 2005), i.e. the temperature dependence of INP number concentrations was not altered. Where different INP parameterisations have been used (Eidhammer et al., 2009; Fan et al., 2010b; Liu et al., 2018; Takeishi and Storelvmo, 2018), the results have in most cases been interpreted in terms of the overall increase in INP number concentration (Fan et al., 2010b; Liu et al., 2018; Takeishi and Storelvmo, 2018). However, there are important structural differences between different INP parameterisations that have not yet been explored in detail. For example, currently available and regularly used parameterisations of INP vary substantially in the dependence of INP activity on temperature. We hypothesise that the difference between parameterisations will be particularly important for deep convective clouds because heterogeneous ice formation occurs over a very wide temperature range from just below 0 to around -38°C in the mixed-phase region of these clouds.  For the same dust particle concentration, predicted INP concentrations can increase by up to three orders of magnitude from -15 to -20°C (corresponding to approximately 1 km altitude change) using an INP parameterisation with a steep temperature dependence (lower INP concentrations at high temperatures and higher INP concentrations at low temperatures) (Atkinson et al., 2013), but by less than one order of magnitude using an INP parameterisation with a shallower dependence (DeMott et al., 2010; Meyers et al., 1992). We hypothesise that such large differences in ice production rates between INP parameterisations are likely to affect cloud properties. In simulations of deep convective clouds over North America (Takeishi and Storelvmo, 2018) there were differences in the magnitude and altitude of droplet depletion depending on INP parameterisation choice (Bigg, 1953; DeMott et al., 2010, 2015).

Uncertainty in mixed-phase cloud properties is compounded further by a lack of quantification of the interaction of heterogeneous freezing with other ice production mechanisms. Ice crystals in the mixed-phase region can also be formed by secondary ice production (SIP) from existing hydrometeors (Field et al., 2017) and droplets can freeze homogeneously below around -33°C (Herbert et al., 2015). In observations of convective clouds with relatively warm cloud-top temperatures (Fridlind et al., 2007; Heymsfield and Willis, 2014; Ladino et al., 2017; Lasher-Trapp et al.,

2016; Lawson et al., 2015), ICNC has frequently exceeded INP number concentrations by several orders of magnitude, suggesting that secondary ice production is the dominant small-ice formation mechanism in mixed-phase regions (Ladino et al., 2017). The importance of heterogeneous ice production relative to secondary and homogeneous freezing has therefore been questioned (Ladino et al., 2017; Phillips et al., 2007) and it has been proposed that INP concentrations may only be relevant up to a threshold needed to initiate SIP (Ladino et al., 2017; Phillips et al., 2007), a value that may be as low as 0.01 $L^{-1}$ (Crawford et al., 2012; Huang et al., 2017) for the Hallett-Mossop process (Hallett and Mossop, 1974). If this is the case, in clouds where SIP may also be initiated by the primary freezing of a few large (~1 mm) droplets in a rising parcel (Field et al., 2017), INP number concentrations may be largely irrelevant to cloud ice properties. The effect of INP and INP parameterisation on convective cloud properties must therefore be examined with consideration for the presence of, and interactions with, SIP.

Here we explore how the choice of INP parameterisation affects the properties of a large and realistic cloud field containing clouds at all levels as well as deep convective systems in the eastern Tropical Atlantic with a focus on the top of atmosphere (TOA) outgoing radiation. The eastern Tropical Atlantic is an ideal location in which to examine the role of INP concentrations in convective cloud systems because, owing to its position at the interface between the Saharan Air Layer and the Inter Tropical Convergence Zone, it is subject to both high levels of convective activity and high loadings of desert dust, a relatively well-defined INP type (DeMott et al., 2003; Niemand et al., 2012; Price et al., 2018). First, we determine how the presence of INP alters the radiative properties of the cloud field. We then examine how the properties of the simulated cloud field, including cloud shortwave reflectivity, cloud fraction and anvil extent, depend on the choice of INP parameterisation. In particular, we examine the importance of the dependence of INP number concentration on temperature, referred to as INP parameterisation slope herein, as a major factor that determines cloud properties. We also examine the effect on cloud properties of the inclusion of SIP due to the Hallett-Mossop process.

# 2. Methods

## 2.1.    Model set-up

**2.1.1 Regional domain and initial conditions**

Simulations described in this article were performed using the Unified Model (UM) version 10.8 (GA6 configuration) (Walters et al., 2017). The UM is a numerical weather prediction model developed by the UK Met Office. We use a regional nest within the global model simulation (Fig. 1a), which has a grid spacing of 1 km (900*700 grid points) and 70 vertical levels. Meteorology of the driving global model is based on operational analysis data. Within the nested domain, the Cloud AeroSol Interacting Microphysics scheme (CASIM) is employed to handle cloud microphysical properties. A global model simulation (UM vn 8.5, GA6 configuration, N512 resolution (Walters et al., 2017)) is used to initialise the nested simulation at 00:00 on the 21st of August 2015 and is used throughout the simulation for the boundary conditions.

The 21st of August 2015 was chosen for simulation to coincide with flight b933 of the Ice in Clouds Experiment – Dust (ICE-D) July-August 2015 field campaign that targeted convective clouds extending to and beyond the freezing level. The aerosol profile measured during flight b933 (Fig. 1b) was used to derive the aerosol profiles prescribed over the nested domain at the beginning of the simulation and are constantly applied at the boundaries. Model profiles were calculated as follows: The UM vn 10.3 was used to simulate a domain comprising the entire Tropical Atlantic and West Africa. This simulation was initiated on the 18[th] August 2015 with a grid spacing of 8 km using the UM operational one-moment microphysics (i.e. not CASIM) and the CLASSIC aerosol scheme with a 6-bin dust model (Johnson et al., 2015a). On the day of the b933 flight (21[st] August 2015), a dust layer was present between 2 and 3 km altitude. Comparison to MODIS AOD data indicates agreement between the model and observations (not shown). This UM vn 10.3 simulation was used to calculate the average dust profile (mass and number concentration) over the CASIM domain on the 21[st] of August 2015 and these dust profiles are applied in the nested domain as the insoluble aerosol profiles (Fig. 1b). The approximate difference between the dust aerosol profile provided by the UM regional simulation and the observed aerosol profile measured during flight b933 (comprising both insoluble and soluble particles) is used as the soluble aerosol profile (Fig. 1b). The simulations are 24 hours in length.

**2.1.2. CASIM microphysics**

CASIM is a multi-moment bulk scheme, which is configured to be two-moment in this work. Both number concentration and mass concentration for each of the five hydrometeor classes (cloud droplets, rain droplets, ice

crystals (or cloud ice), graupel, snow) are prognostic variables. The model set-up is very similar to that used in
Miltenberger et al. (2018) including the parameter choices within CASIM. CASIM has been used and tested previously
in simulations of coastal mixed-phase convective clouds (Miltenberger et al., 2018), South-East Pacific stratocumulus
clouds (Grosvenor et al., 2017), Southern Ocean supercooled shallow cumulus (Vergara-Temprado et al., 2018),
midlatitude cyclones (McCoy et al., 2018) and CCN-limited Arctic clouds (Stevens et al., 2018). The parameters used in
the representation of the size distribution, density and terminal fall speed velocities of each of the five hydrometeor
classes represented by CASIM are shown in Table 2 of Miltenberger et al. (2018).
Cloud droplet activation is parameterised according to (Abdul-Razzak and Ghan, 2000). The soluble accumulation
mode aerosol profile shown in Fig. 1b is used for cloud droplet activation and a simplistic CCN activation
parameterisation is included for the insoluble aerosol mode(Abdul-Razzak and Ghan, 2000) that assumes a 5% soluble
fraction on dust. Scavenging of CCN or INP is not represented. Collision-coalescence, riming of ice crystals to graupel
and aggregation of ice crystals to snow is represented. Rain drop freezing is described using the parameterisation of
Bigg (1953). For reference, the modelled domain-mean out-of-cloud temperature and relative humidity are shown in
Fig. 1c. The model time-step is 5 seconds.
Heterogeneous ice nucleation is represented using 5 different parameterisations: Cooper (1986) (C86), Meyers et al.
(1992) (M92), DeMott et al. (2010b) (D10), Niemand et al. (2012) (N12) and Atkinson et al. (2013) (A13) (Fig. 2). C86
and M92 calculate a freezing rate based on temperature and are independent of aerosol concentration. D10
calculates an INP concentration from temperature and the concentration of insoluble dust aerosol with a diameter
greater than 0.5 μm. N12 and A13 calculate an INP concentration from the temperature dependent active surface site
density and the surface area of insoluble dust aerosol ($n_s$). For A13, a potassium-feldspar fraction of 0.25 is assumed.
This is the upper recommended fraction (Atkinson et al., 2013) which was deemed appropriate because of the study
region's exposure to Saharan dust outflow. M92 is described as a deposition and condensation freezing
parameterisation (Meyers et al., 1992) and is often used alongside an immersion freezing parameterisation in
modelling studies (Deng et al., 2018; Fan et al., 2010b, 2010a; Gibbons et al., 2018). However, the M92
parameterisation is based on aircraft continuous flow diffusion chamber measurements and those measurements
should capture all relevant nucleation mechanisms (see Vali et al., 2015). To represent nucleation at conditions
relevant for clouds with liquid water present, we have set the saturation term in the M92 parameterisation to water
saturation.  One simulation is conducted with no active heterogeneous ice nucleation representation (NoINP). The INP
parameterisations inspect the conditions (temperature, cloud droplet number, ICNC) and aerosol concentrations
within a gridbox and use that information to predict an ice production rate via heterogeneous freezing. The
supercooled droplets are depleted by the freezing parameterisation, but scavenging of INPs is not represented.
Homogeneous freezing of cloud droplets is parameterised according to Jeffery and Austin (1997).
The INP parameterisations tested in this study represent only immersion freezing. Heterogeneous ice nucleation by
deposition and contact nucleation are not represented. Other mechanisms of heterogeneous ice formation should be
tested and included in future studies but was beyond the scope of this work. However, immersion freezing is expected
to be the dominant mechanism of heterogeneous ice formation in convective clouds (Ansmann et al., 2008; De Boer
et al., 2011; Kanji et al., 2017) and therefore the simulations presented here should capture the majority of
heterogeneous ice nucleation relevant for cloud properties. Immersion and homogeneous freezing of haze droplets
are not represented, but it is unlikely that they contribute significantly to ice crystal number concentration in the main
anvil cloud derived from mixed-phase cloud regions. However, the importance of these mechanisms on anvil cloud
properties should be investigated in future work.Secondary ice production (SIP) is represented using an approximation
of the Hallett-Mossop process which occurs between -2.5 and -7.5°C. The efficiency of the Hallett-Mossop process
increases from -2.5 and -7.5°C to 100% at -5°C.  The rate of splinter production per rimed mass is prescribed with 350
new ice splinters produced per milligram of rime at -5°C. Splinters are produced from rime mass of snow and graupel.
The ice splinters produced by the representation of the Hallett-Mossop process are the smallest allowable size of ice
in the model (i.e. $10^{-18}$ kg, volume radius ~0.11 μm). The rate of splinter production by the Hallett-Mossop process is
based on the best available estimate of the efficacy of the mechanism (Connolly et al., 2006; Hallett and Mossop,
1974; Mossop, 1985). In-situ cloud observations have frequently observed ICNC that could be explained by the Hallett-
Mossop process, but the mechanism underlying the Hallett-Mossop process as well as the ice particle production rate
remain uncertain and not well quantified (Field et al., 2017). A maximum splinter production rate of 350 per milligram
of rimed material has been measured in a number of laboratory studies (Hallett and Mossop, 1974; Mossop, 1985)
and has been applied as the best estimate here and in previous modelling studies (Connolly et al., 2006), although
other rates have also been measured (Heymsfield and Mossop, 1984; Saunders and Hosseini, 2001). Uncertainties
regarding the rate of splinter production by Hallett-Mossop are an important consideration that will be investigated in
future work; this study explores the structural uncertainty of the presence/absence of the Hallett-Mossop process as
currently understood. Other mechanisms of SIP such as collision fragmentation, droplet shattering and sublimation
fragmentation have been proposed (Field et al., 2017), but are not represented in these simulations, in part because
they are very poorly defined and it is not clear how important they are.  Other studies have attempted to model some
of these additional SIP processes (Phillips et al., 2018; Sullivan et al., 2018) but that was beyond the scope of this
study.

### 2.1.3. Cloud radiation

The radiative processes are represented by the Suite of Community RAdiative Transfer codes based on Edwards and
Slingo (SOCRATES) (Edwards and Slingo, 1996; Manners et al., 2017), which considers cloud droplet number and mass,
as well as ice crystal and snow water paths for the calculation of cloud radiative properties. It does not explicitly
consider changes in ice crystal or snow number concentration or size (though changes in number and size will affect
mass concentrations which are considered), and does not consider any changes to rain or graupel species. The cloud
droplet single scattering properties are calculated from the cloud droplet mass and effective radius in each gridbox
using the equations detailed in Edwards and Slingo (1996). Snow and ice are combined to form one ice category for
the purposes of the radiation calculations. The single scattering properties of this snow and ice category are calculated
from their combined mass and the ambient temperature. The parameterisation of bulk optical properties of snow and
ice used in the model is detailed in Baran et al. (2014).
The radiative properties (shortwave, longwave and total radiation) are calculated for daylight hours only, i.e. 10:00-
17:00 UTC. For all other modelled properties presented, except when plotted against a corresponding radiative
property, values are calculated for the last 14 hours of the simulation, i.e. from 10:00 - 24:00. The sensitivity of the
outgoing longwave radiation and the cloud fraction to time period selection was tested and found to have little
impact. The overall outgoing radiation (shortwave + longwave) will be sensitive to the time period selection owing to
the absence of outgoing shortwave radiation at night-time. We focus on the radiation during daylight hours only
because our simulation is only 24 hours in length owing to computational restrictions and therefore when the spin-up
period is excluded from the analysis, less than 24 hours of simulation data remains with much of the night-time hours
removed with the spin-up period.
Changes to outgoing radiation from cloudy regions and changes in cloud fraction both contribute to the total overall
change in outgoing radiation between two simulations. The contributions from changes in outgoing radiation from
cloudy regions and cloud fraction to the overall radiative differences between simulations were calculated separately
as described below. The cloudy regions contribution, i.e. the difference in outgoing radiation between two cloudy
regions due to changes in cloud albedo or thickness ignoring any changes in cloud fraction, $(\Delta Rad_{REFL})$ to a

domain radiative difference between a sensitivity simulation (s) and a reference simulation (r) (s – r) is calculated using Eq. (1).

$$\Delta Rad_{REFL} = cf_r \times \Delta Rad_{cl} \qquad (1)$$

where $cf_r$ is the cloud fraction of simulation r and $\Delta Rad_{cl}$ is the change in outgoing radiation from cloudy areas only between simulations (s – r). The reference run (r) in Sections 3.1 – 3.4 refers to the NoINP simulation while the sensitivity run (s) are simulations which include an INP parameterisation. In Section 3.5, the reference run (r) refers to a simulation which has no representation of SIP and the sensitivity run (s) to a simulation which includes SIP due to the Hallett-Mossop process. The contribution of cloud fraction changes, i.e. the change in radiation that can be attributed to an area of clear sky in simulation s becoming cloudy in simulation r or vice versa, to the total change in domain outgoing radiation ($\Delta Rad_{CF}$) is calculated using Eq. (2).

$$\Delta Rad_{CF} = \left(Rad_{r,cl} - Rad_{r,cs}\right) \times \Delta cf \qquad (2)$$

Where $Rad_{r,cl}$ is the mean outgoing radiation from cloudy regions in simulation r and $Rad_{r,cs}$ is the mean outgoing radiation from clear sky regions in simulation r and $\Delta cf$ is the difference in domain cloud fraction between simulations s and r (s-r). There is interaction between the outgoing radiation from cloudy regions and cloud fraction changes ($\Delta Rad_{INT}$) which is calculated in Eq. (3).

$$\Delta Rad_{INT} = \Delta Rad_{cl} \times \Delta cf \qquad (3)$$

The contribution of changes in the outgoing radiation from clear sky areas ($\Delta Rad_{CSKY}$) can be calculated as shown in Eq. (4).

$$\Delta Rad_{CSKY} = \Delta Rad_{cs} \times (1 - cf_s) \qquad (4)$$

Where $\Delta Rad_{cs}$ is the change in mean outgoing radiation from clear sky areas between simulations s and r and $cf_s$ is the cloud fraction of simulation s.

The total outgoing radiation difference between simulations s and r ($\Delta Rad_{s-r}$) is therefore as shown in Eq. (5).

$$\Delta Rad_{s-r} = Rad_s - Rad_r = \Delta Rad_{REFL} + \Delta Rad_{CF} + \Delta Rad_{INT} + \Delta Rad_{CSKY} \qquad (5)$$

The interaction term $\Delta Rad_{INT}$ and the clear sky term ($\Delta Rad_{CSKY}$) were found to be negligible and are therefore
ignored for the purposes of this paper.
**2.1.4. Model simulations**
The conducted simulations are as follows:

-    Five simulations with different heterogeneous ice nucleation parameterisations (C86, M92, D10, N12 and

A13) with a representation of the Hallett-Mossop process (SIP_active).

-    One simulation with no heterogeneous ice nucleation (NoINP), but with a representation of the Hallett-

Mossop process (SIP_active).

-    Five simulations with different heterogeneous ice nucleation parameterisations (C86, M92, D10, N12 and

A13) without a representation of the Hallett-Mossop process (SIP_inactive).

The INP number concentration ([INP]) predicted by the five INP parameterisations (C86, M92, D10, N12, A13) are
compared with the available measurements from the study region (Price et al., 2018; Welti et al., 2018) in Fig. 2,
including those taken during the ICE-D field campaign (Price et al., 2018). All parameterisations are in reasonable
agreement with the measurements (and with each other) at around -17°C, but deviate strongly at higher and lower
temperatures. It should be noted that all parameterisations tested in this work were developed between specific
temperature ranges and extrapolation beyond these temperatures adds uncertainty. However, for the purposes of
this paper and to allow a direct comparison between parameterisations, all parameterisations have been applied
between 0 and -37°C. Importantly, the INP parameterisation slopes of the chosen parameterisations span the range
used within regional models (from a shallow $d\log_{10}[INP]/dT = -0.07$ in M92 (Meyers et al., 1992) to a steep
$d\log_{10}[INP]/dT = -0.45$ in A13 (Atkinson et al., 2013)).
When analysing the simulation output, cloudy grid boxes were classed as those containing more than $10^{-5}$ kg kg$^{-1}$
condensed water from cloud droplets, ice crystals, graupel and snow. Rain was not included to ensure analysis did not
include areas below cloud base. Other cloud thresholds were tested and found to have no notable effect on the
results. For cloud categorisation into low, mid and high clouds, model vertical columns containing cloudy grid boxes
were categorised by cloud altitude. Low cloud occurs below 4km, mid cloud between 4 and 9 km and high cloud above
9 km. Columns with cloudy grid boxes in two or more cloud categories were classified as mixed category columns
according to the vertical placement of the cloudy grid boxes, e.g. low/high for columns containing cloud below 4 km

and above 9 km. 4 and 9 km were chosen as the low/mid and mid/high division points because they are just below

two well-defined peaks in cloud base heights (not shown) and roughly correspond to the beginning of the

heterogeneous and homogeneous freezing regions, respectively. For the correlation analysis where model outputs

were plotted against parameterisation slope (dlog10[INP]/dT), a straight line was fitted to the D10 parameterisation

between -3 and -37°C to obtain an approximate INP parameterisation slope. Other temperature ranges were tested

and were found to have no notable effect on results.

## 2.2.    The observed case

MODIS visible images of the 21$^{st}$ August 2015 are shown in Fig. 3 (a, b) alongside snapshots of the TOA outgoing

longwave radiation in one of our simulations (c, d). The simulated cloud field has more cloud-free areas than the

satellite images but in general produces clouds similar to those shown in the satellite image and in approximately the

correct location. Overall the simulations produce a complex and realistic cloud field. Snapshots of the simulated model

TOA outgoing shortwave radiation are shown in Fig. A1.

In-situ measurements of cloud and aerosol properties were made using the UK FAAM Bae-146 research aircraft, which

was flown from Praia, Cape Verde Islands. An extensive suite of in-situ aerosol and cloud particle instruments were

operated onboard the aircraft and are described in detail in Lloyd et al. (2019). The aircraft penetrated the growing

convective clouds at a range of altitudes from just below the freezing level up to -20°C.  In order to show that the

model reproduces the observed conditions, the observational data were compared to the conditions in modelled

clouds of similar size to those the aircraft flew in (10 – 150 km$^2$) where a comparison was thought appropriate.

Comparisons of a selection of simulated cloud properties with aircraft data are shown in Fig. A2. In-cloud

measurements from the aircraft were selected using the same total water content threshold as for the model data

(10$^{-5}$ kg kg$^{-1}$). Note that observational data only samples clouds along the 1D flight path, while model results include all

grid points inside the selected clouds.

The vertical wind and cloud droplet and ice number concentrations are shown Fig. A2. The vertical wind speeds from

the model and aircraft measurements agree well (Fig. A2a). The aircraft data exhibit less measurements of vertical

wind speeds above 10 m s$^{-1}$ but that is expected since the aircraft was purposefully not flown in very high updraft

speeds. The aircraft cloud droplet number concentration (CDNC), measured using a Droplet Measurement Technique

(DMT) cloud droplet probe (which allows measurement of the cloud droplet size distribution for particles with

diameters between  3 and 50 μm (Lloyd et al., 2020)), falls predominantly in the regions of parameter space most

highly populated by model data when plotted against vertical wind speed (Fig. A2b). Note that the simulated points in

Figure A1b represent values of CDNC and updraft speed in all cloudy gridboxes, not just those at cloud base. The

updraft speed is collocated with CDNC and therefore does not necessarily represent the updraft speed at which the

cloud droplets were activated. The higher CDNC values exhibited in the model data may be due to the higher updraft

speeds which were not measured by the aircraft. The observed ICNC was derived from measurements using the DMT

Cloud Imaging Probes (CIP-15 and CIP-100, photodetector widths of 15 and 100 μm respectively, both with 64

detector elements) and the SPEC Stereoscopic optical array probe covering a size range from 10 to 6200 μm using the

SODA2 (System for OAP (optical array probe) Data Analysis) processing code (McFarquhar et al., 2017) to reconstruct

ice particle images that are fully contained within the probe sample volume. Because of uncertainties in the optical

array probe sample volume for very small images, only ice particle images greater than 100 μm were included. The

aircraft ICNC fall almost entirely within the range of the model values (Fig. A2c).

# 3. Results

## 3.1.    Effect of INP and INP parameterisation on outgoing radiation

We first examine the effect of INP parameterisation on the TOA outgoing daytime (10:00-17:00 UTC) radiation relative

to the simulation where the only source of primary ice production was through homogeneous freezing (NoINP). Ice

crystals formed via homogeneous freezing and sedimented to lower levels, can initiate ice production via the Hallett-

Mossop process once converted to snow or graupel. When contrasting the effect of different INP parameterisations in

Sect 3.1-3.4, the Hallett-Mossop process was always active including in the NoINP simulation. As stated in Sect. 2.1.3,

the radiation code is represented by the Suite Of Community RAdiative Transfer codes based on Edwards and Slingo

(SOCRATES) (Edwards and Slingo, 1996; Manners et al., 2017), and responds to changes in cloud droplet number and

cloud droplet, ice crystal and snow mass. The results detailed below relate to either the domain-wide properties or all

in-cloud regions within the domain. This means that the results describe the direct and indirect changes, for example

changes to the Hallett-Mossop ice production, occurring due to the presence of INP across all cloud present in the

domain, including low-level liquid clouds, mixed-phase clouds without a convective anvil and very deep convective

clouds with an anvil. The effects of INP parameterisation and SIP on convective anvils are discussed in Sect. 3.4.

Domain-mean TOA outgoing radiation (daylight hours, shortwave plus longwave) is enhanced by the inclusion of INP in all cases (Fig. 4a). The enhancement in outgoing radiation varies between 2.6 W m$^{-2}$ for D10 and 20.8 W m$^{-2}$ for A13 relative to the NoINP simulation. There is a variation of up to 18.2 W m$^{-2}$ depending on the chosen representation of heterogeneous ice nucleation, which shows that the INP parameterisation can affect outgoing radiation as much as excluding or including heterogeneous freezing altogether. The difference in radiation between the NoINP and the simulations where INP are present are caused mainly by changes to outgoing shortwave radiation. The inclusion of INP enhances outgoing shortwave radiation by between 5.3 W m$^{-2}$ for D10 and 26.6 W m$^{-2}$ for A13 (Fig. A3a). Differences in outgoing longwave radiation are comparatively small (-2.7 W m$^{-2}$ for D10 to -5.8 W m$^{-2}$ for A13; Fig. A3b) due to similar cloud top heights between simulations of these thermodynamically limited clouds. Bear in mind that SIP was active (SIP_active) in the simulations summarised in Fig. 4a, including in the NoINP simulation in which the Hallett-Mossop process can be initiated by settling ice-phase hydrometeors (either by settling homogeneously frozen ice crystals subsequently converted to snow or graupel, or by settling snow or graupel formed from homogeneously frozen ice crystals at upper cloud levels), indicating that these cloud systems are sensitive to INP even in the presence of SIP. This is consistent with a comparatively small change in TOA radiation when SIP is active relative to when it is inactive (Fig. 4b and 3c) (we discuss the role of SIP in more detail in Sect. 3.5).

The slope of the INP parameterisation (i.e. the dependence of INP number concentration on temperature) is a key determinant of the outgoing radiation. There is a statistically significant correlation between INP parameterisation slope and total TOA outgoing radiation ($r^2$ = 0.75, $p$ < 0.01, $n$ = 10) (Fig. 4c). Changes in outgoing radiation due to the presence of INP are caused by a combination of changes to the outgoing radiation from cloudy regions, caused by changes in cloud structure and microphysical properties, and changes to domain cloud fraction, whose contributions to the total radiative difference are shown in Fig. 4a (left and centre). In order to appreciate the reasons for these trends, we will now take a closer look at the effect of INP on outgoing radiation from cloudy regions only, domain cloud fraction and cloud type.

### 3.2. Effect of INP and INP parameterisation on outgoing radiation from cloudy regions

Here we discuss the changes in daytime outgoing radiation from cloudy regions only due to INP parameterisation choice. Daytime outgoing radiation from cloudy regions increases due to INP for all but one INP parameterisation (Fig. 5a). The absolute change in outgoing radiation from cloudy regions is between –0.8 (D10) and +28.1 (A13) W m$^{-2}$, and the larger values are a result of large increases in reflected shortwave (up to +37.2 W m$^{-2}$) and relatively moderate decreases in outgoing longwave radiation (up to –11.1 W m$^{-2}$) from cloudy regions. The above absolute changes in

outgoing radiation from cloudy regions contribute between −0.7 and +11.4 W m$^{-2}$ to the domain-mean change in
outgoing radiation due to the presence of INP (Fig. 4a, cloudy regions contribution).
The enhancement of outgoing radiation from cloudy regions due to INP is caused primarily by increases in cloud
condensate relative to the NoINP simulation (Fig. 5b). When INP are included in a simulation, snow and cloud droplet
water path are enhanced, causing increases in total cloud condensate, despite decreases (in all except A13) in ice
crystal water path due to a reduction in ice crystal number and mass concentrations caused by a reduction in the
availability of cloud droplets for homogeneous freezing. Snow, cloud droplets and ice crystals are the hydrometeors
that affect outgoing radiation in CASIM and the combined water path of these three species is significantly positively
correlated with cloud shortwave reflectivity ($r^2$ = 0.62, $p$ < 0.01, $n$ = 11) (Fig. 5c). The mechanism for this INP induced
increase in cloud condensate and consequently cloud shortwave reflectivity is as follows: When heterogeneous ice
nucleation is active, liquid is consumed in the warmer regions of mixed-phase clouds because of increased
heterogeneous ice nucleation (Fig. 2) and SIP (Fig. A4a). The resultant additional ice crystals in mixed-phase regions
facilitate riming causing increases in snow and graupel (Fig. A4c, d), increasing snow water path and reflectivity in
mixed-phase and ice clouds. At the same time, the enhanced production of relatively heavy snow and graupel
increases precipitation which on melting to form rain below the freezing level and subsequent evaporation below 4
km, reduces out-of-cloud temperature and increases relative humidity (Fig. A5a, b). This leads to increases in water
path in low-level liquid clouds and thus an enhancement in their shortwave reflectivity.
However, increases in total cloud condensate alone cannot account for the differences in outgoing radiation from
cloudy regions between simulations using different INP parameterisations, which are caused by a combination of
cloud microphysical responses. We find that outgoing radiation from cloudy regions is significantly negatively
correlated with INP parameterisation slope ($r^2$ = 0.63, $p$ < 0.01, $n$ = 10) (Fig. 6a), i.e. simulations using a steep INP
parameterisation have a higher outgoing radiation from cloudy regions. This result makes sense when we consider the
relationships between INP parameterisation slope and a multitude of cloud microphysical properties affecting cloud
radiative properties. In particular, a steep INP parameterisation results in a mixed-phase cloud region characterised by
a higher ice crystal water path aloft ($r^2$ = 0.80, $p$ < 0.01, $n$ = 10; Fig. 6b) and higher cloud droplet number
concentrations at the bottom of the mixed-phase region ($r^2$ = 0.89, $p$ < 0.01, $n$ = 10; Fig. 6c) when compared to
shallower parameterisations. A steeper INP parameterisation slope allows increased transport of liquid to upper cloud
levels due to lower rates of heterogeneous freezing at the mid-bottom region of the mixed-phase cloud (lower
supercooling, Fig. 2) and SIP at high temperatures (Fig. A4a). This, combined with higher INP concentrations at low
temperatures (Fig. 2), increases ICNC at upper mixed-phase altitudes, as well as enhancing the lifetime of liquid cloud
droplets at lower altitudes in the mixed-phase region when compared to shallower INP parameterisations.

### 3.3. Effect of INP and INP parameterisation on cloud fraction

Overall cloud fraction is increased by INP for all INP parameterisations and these increases in cloud fraction contribute
about as much to changes in domain-mean daytime radiation as the changes in outgoing radiation from cloudy
regions (Fig. 4a, cloud fraction contribution). Increases in domain cloud fraction due to INP are driven by cloud cover
increases in the warm and mixed-phase regions of the cloud (~ 4 -6 km), offset somewhat by decreases in the cloud
fraction due to reduced homogeneous freezing in the ~ 10 - 14 km regime (Fig. 7a). Cloud fraction increases at mid-
levels occur because heterogeneous ice nucleation induces an increase in precipitation-sized particles (snow and
graupel) which sediment to lower levels and moisten the atmosphere by evaporation (Fig. A5a, b). This increases new
cloud formation and may prolong the lifetime of existing cloud cells. Additionally, increased droplet freezing and
riming in the mixed-phase cloud region releases latent heat and invigorates cloud development with increases in
updraft speed just above 4 km (Fig. A5c). The increased cloud fraction at mid-levels due to INP are partially offset by a
reduced cloud fraction above 10 km (Fig. 7a) which is caused by an INP induced enhancement in freezing and riming in
the mixed-phase region reducing moisture transport to the homogeneous freezing regime. The ability of
heterogeneous freezing to reduce the availability of moisture for homogeneous freezing has been previously observed
(e.g. Gasparini et al., 2020; van den Heever et al., 2006; Kärcher and U. Lohmann, 2003; Lohmann and Gasparini, 2017;
Phillips et al., 2005, 2007; Storelvmo et al., 2013).
The effects of INP on the altitude profile of cloud fraction are strongest for shallow INP parameterisation slopes, which
have a freezing profile most different to that of the NoINP simulation (Fig. 7a). At 5 km, the shallowest
parameterisation (M92) causes the largest increase in cloud fraction, while the steepest parameterisation (A13)
causes the smallest ($r^2$ = 0.83, $p < 0.05$, $n$ = 5). At 12 km, the order is reversed, and steep parameterisations exhibit the
highest cloud fraction ($r^2$ = 0.94, $p < 0.01$, $n$ = 5). The largest cloud fraction-induced increases in outgoing radiation
relative to the NoINP simulation (Fig. 4a) are seen in simulations using steeper INP parameterisations because these
simulations exhibit higher cloud fractions at high altitudes (~12 km), translating into the higher total cloud fraction.
These slope dependent changes in cloud fraction are explained by a relationship between cloud fraction and several
microphysical properties affecting cloud fraction. For example, steeper INP parameterisations produce higher ICNC at
the top of the mixed-phase region (10 km) as well as higher ratios of ice crystal mass to snow and graupel mass within
the homogeneous freezing region (12 km) (Fig. 7b, c). A higher number and mass of ice crystals relative to those of
larger precipitation-sized hydrometeors with the steepest parameterisations results in lower frozen hydrometeor
sedimentation, a longer cloud lifetime and a higher cloud fraction.

### 3.4.    Effect of INP and INP parameterisation on cirrus anvils

Our results show that the INP parameterisation affects the properties and spatial extent of cirrus anvils. We define
cirrus anvils to be regions where cloud is present above 9 km only (further details available in Sect. 2.1.4). 2D aerial
images of cloud categorisation (Fig. 8a-f) show well-defined regions of anvil cloud (light blue - H) surrounding a large
convective system containing clouds at a range of altitudes from <4 km to >9 km. There are clearly differences in the
extent and position of cloud categories between simulations (Fig. 8a -f).
The presence of INP reduces convective anvil extent by between 2.1 and 4.1% of the domain area depending on the
choice of INP parameterisation (Fig. 8 g), corresponding to a decrease in anvil cloud of between 22 and 53% relative to
the NoINP simulation (not shown). The reduction in anvil extent in the presence of INP is caused by increased liquid
consumption at all mixed-phase levels, due to heterogeneous freezing, enhanced SIP and increased graupel and snow
production, reducing the availability of cloud droplets for homogeneous freezing (Fig. A4b), reducing ICNC at cloud-
top, and reducing cloud anvil extent (Fig. 8g), in agreement with previous studies  (e.g. Gasparini et al., 2020; van den
Heever et al., 2006; Kärcher and U. Lohmann, 2003; Lohmann and Gasparini, 2017; Phillips et al., 2005, 2007;
Storelvmo et al., 2013).
Reductions in anvil extent caused by INP are somewhat offset by the overall increases in cloud fraction across the
domain (Fig. 8g). However, it is possible that the effect of INP and INP parameterisation choice on anvil cloud fraction,
and the contribution of anvil cloud to overall cloud fraction and radiative changes, would become larger with a longer
analysis period. This is because detrained convective anvils can persist longer in the atmosphere than the convective
core that creates them (Luo and Rossow, 2004; Mace et al., 2006), but this is beyond the scope of the current study.

### 3.5.    Importance of secondary ice production

It has been argued that the observed (or derived) primary ice particle production rate is unimportant for convective
cloud properties when secondary ice production (SIP) is active (Fridlind et al., 2007; Heymsfield and Willis, 2014;
Ladino et al., 2017; Lawson et al., 2015) because primary ice crystal concentrations are often overwhelmed by ice
crystals formed via SIP (Field et al., 2017). However, the results shown in Fig. 4a (in which the simulations included
SIP) do not support this argument. We find that the microphysical and radiative properties of the cloud field depend
strongly on the properties of the INP even when SIP due to the Hallett-Mossop process occurs. Furthermore, the

effect of including SIP on daylight domain-mean outgoing radiation varies between −2.0 W m$^{-2}$ and +6.6 W m$^{-2}$ (Fig.

4b), showing that the presence of the Hallett-Mossop process has a smaller effect than the INP parameterisation and

that the sign and magnitude of this effect depends on the INP parameterisation. The mean effect on daylight domain-

mean outgoing radiation of including INP is +9.8 W m$^{-2}$ whereas the mean effect of including SIP via the Hallett-

Mossop process is +2.9 W m$^{-2}$. Therefore, rather than primary ice being simply overwhelmed by SIP, it actually

determines how SIP affects cloud microphysics. Other mechanisms of SIP have been proposed (Field et al., 2017;

Korolev and Leisner, 2020; Lauber et al., 2018) and the impact of INP on cloud properties in the presence of these

mechanisms, particularly those present at temperatures below 10°C such as droplet shattering (Lauber et al., 2018),

should be tested in future but this was beyond the scope of the present study.

The effect of SIP on the radiative properties of the cloud field is dependent on INP parameterisation choice, both in

magnitude and sign of change (Fig. 4b). SIP makes the clouds more reflective independent of the chosen

parameterisation (Fig. 4b, cloudy regions contribution) due to increases in snow and cloud droplet water path. N12

and A13 have the largest overall radiative response to SIP because changes to the radiative forcing from cloudy

regions and cloud fraction contributions act to increase outgoing radiation (Fig. 4b). However, the cloud fraction

response to SIP is opposite for C86, M92 and D10 meaning the cloudy regions and cloud fraction contributions act in

opposite directions, reducing the total radiative forcing.

The different response of the domain cloud fraction to the presence of SIP is caused by substantial variation between

simulations in the anvil cloud extent (Fig. 8h), from an increase of 10% (+0.9% of the domain area) in N12 to a

decrease of 40% (-3.6% of the domain area) in M92 (Fig. 8h). These non-uniform changes in cloud fraction and

outgoing radiation can be explained by differences in the response of cloud freezing profiles to SIP due to variations in

INP parameterisation slope. For all INP parameterisations, SIP reduces the availability of liquid at higher altitudes. For

shallower parameterisations such as M92 this causes a reduction in the amount of cloud droplets reaching the

homogeneous freezing regime and thereby reduces ICNC and cloud anvil spatial extent. However, in simulations using

a steep parameterisation, almost all available droplets are frozen heterogeneously before they reach the

homogeneous regime (see reduced homogeneous ice production rates in N12 and A13 in Fig. A4b). Therefore, in

simulations using a steeper parameterisation, such as N12, a reduction in liquid availability due to SIP occurs at the

top of the heterogeneous freezing regime, reducing the availability of liquid for riming, causing a reduction in frozen

hydrometeor size at high altitudes, a reduction in hydrometeor sedimentation and an increase in anvil extent. The

effects of INP parameterisation slope and the Hallett-Mossop process on the simulated cloud field properties are

summarised in Fig. 9. Overall, our simulations show that INP parameterisation choice and slope is an important determinant of cloud field micro- and macrophysical properties, even when SIP is active, and that choice of INP parameterisation affects the cloud field response to SIP.

# 4. Limitations of this modelling study

The lack of consideration of ice and snow particle number by the SOCRATES radiation scheme is an important limitation of the results presented here. Changes to ICNC, without a co-occurring change in ice crystal mass concentrations, will not be reflected in modelled radiative fluxes. However, our results are still very relevant for climate model simulations as climate models do not typically account for ICNC in their radiation calculations and have frequently been shown to poorly represent ice crystal mass concentrations (Baran et al., 2014; Waliser et al., 2009). The SOCRATES representation of radiation with a dependence on ice mass is a more accurate and realistic representation of radiation than is seen in many climate models which often derive bulk optical properties using empirically derived deterministic relationships between ice particle size and environmental temperature and/or ice water content (Baran et al., 2014; Edwards et al., 2007; Fu et al., 1999; Gu et al., 2011). However, the effect of INP parameterisation on deep convective clouds radiative properties using a radiation code that considers ice particle number should be explored in future studies. The sensitivity of the cloud field to the chosen INP parameterisation and SIP indicates the importance of accurately representing ice water content in climate models and linking this ice water content to ice-nucleating particle type.

Another limitation of the SOCRATES radiation code is its lack of consideration of rain and graupel particles. The effects of these hydrometeors are expected to be less than that of ice, snow and cloud droplets as they precipitate faster and therefore have a shorter lifetime. Furthermore, the effect of graupel on the tropical longwave radiative effect has been found to be negligible and dwarfed by that of snow (Chen et al., 2018). The global radiative effect of rain has also been found to be small in the vast majority of cases even at high temporal and spatial resolution (Hill et al., 2018). The effect of the incorporation of these hydrometeors into radiative transfer parameterisations should however be tested in future studies.

The use of both aerosol-dependent (D10, N12, A13) and solely-temperature dependent (C86, M92) parameterisations in this study means that we have examined the radiative sensitivity of a complex cloud field to a larger variety of INP

parameterisations used in weather and climate models than if we had exclusively used parameterisations that
consider aerosol concentration. However, this experimental design has limitations. For example, due to the lack of
aerosol dependence of the C86 and M92 schemes a 'presumed 'dust concentration is implicitly present in these two
cases and remains uniform throughout the simulation period. The effect of INP parameterisation choice on convective
cloud field properties should also be examined with the inclusion of aerosol scavenging but this was beyond the scope
of this study. Aerosol scavenging would allow the aerosol number concentration to be reduced by cloud droplet
activation and the number of dust particles within cloud droplets to be tracked and depleted when frozen
heterogeneously. In the simulations presented here, the heterogeneous freezing rate is calculated using the
interstitial aerosol number concentration and the ICNC of the gridbox in question meaning that ice crystals advected
into the gridbox will reduce the heterogeneous nucleation rate even if they were frozen elsewhere in the domain.
Furthermore, while many cloud macro- and microphysical were correlated with INP parameterisation slope, the slope
of the parameterisation at low temperatures for the A13 and N12 parameterisations can be flat because the
parameterisations plateau once they reach the number concentration of dust represented in the model gridbox in
question. This means that at high dust concentrations, the slope of the INP parameterisation correlates with the INP
concentration at temperatures between -25 and -35°C (Figure 2). This means that the absolute number concentration
of aerosols capable of nucleating ice is not decoupled from the INP parameterisation slope in some INP
parameterisations and that some cloud responses attributed to changes in the INP parameterisation slope may have
in fact been caused by the absolute INP number concentration at cold temperatures. The relative importance of the
INP parameterisation slope and the absolute number concentration of aerosols capable of nucleating ice will be
investigated in future work. However, whether the INP number concentration plateaus at cold temperatures is
determined in part by the INP parameterisation slope, and correlations with INP parameterisation slope are evident at
both warm and cold cloud altitudes indicating the importance of the INP parameterisation slope.
This study utilised our best estimate of ice production by the Hallett-Mossop process (Connolly et al., 2006; Hallett
and Mossop, 1974; Mossop, 1985), the most well-studied SIP mechanism, to try and understand the effect of the
process, as currently understood, on deep convective cloud properties. The work indicates that INP concentrations at
all mixed phase temperatures can be important for cloud properties even in the presence of the Hallett-Mossop
process, and that the impact of the Hallett-Mossop process depends on INP number concentrations. The dependence
of the rate of ice production by the Hallett-Mossop process on INP number concentrations (Figure A5a) in particular
highlights that the role of SIP in clouds may be dependent on INP. However, the rate of ice production by the Hallett-
Mossop process is very uncertain and other mechanisms of SIP have also been proposed (Field et al., 2017). We
recommend that similar studies examining the effect of INP should be conducted with the inclusion of other proposed
SIP mechanisms, particularly those that may be present at temperatures below -10°C, such as droplet shattering
(Lauber et al., 2018). However, this was beyond the scope of the present study due in part to the lack of quantification
and parameterisations for these other mechanisms (Field et al., 2017). Future work will attempt to overcome the
above caveats by using statistical emulation (Johnson et al., 2015b) to examine the interacting effects of dust number
concentration, INP parameterisation slope and SIP in an idealised deep convective cloud.

## 5. Conclusions

We quantified the effect of INP parameterisation choice on the radiative properties of a deep convective cloud field
using a regional model with advanced double-moment capabilities. The simulated domain exceeds 600,000 km$^2$ and
therefore captures the effects of INP and INP parameterisation on a typical large, complex and heterogeneous
convective cloud field. The presence of INP increases domain-mean daylight TOA outgoing radiation by between 2.6
and 20.8 W m$^{-2}$ and the choice of INP parameterisation can have as large an effect on cloud field properties as the
inclusion or exclusion of INP. These effects are evident even in the presence of SIP due to the Hallett-Mossop process,
refuting the hypothesis that INP is irrelevant beyond a minimum concentration needed to initiate the Hallett-Mossop
process (Crawford et al., 2012; Ladino et al., 2017; Phillips et al., 2007). An important caveat of this result is that other
SIP mechanisms, such as droplet shattering (Ladino et al., 2017; Lauber et al., 2018), are not represented in our model
simulations. Furthermore, the effects of SIP on the cloud field properties are strongly dependent on INP
parameterisation choice. Both the magnitude and direction of change in cloud fraction and total outgoing radiation
due to SIP varies according to INP parameterisation choice. Microphysical alterations to cloud properties are
important contributors to radiative differences between simulations, in agreement with previous studies documenting
the effect of aerosol-cloud interactions to the radiative forcing by deep convective clouds (Fan et al., 2013). For
example, increasing cloud condensation nuclei concentrations, with no perturbations to INP, was shown to increase
cloud albedo and cloud fraction, deepen clouds and increase TOA outgoing radiation by 2-4 W m$^{-2}$ (Fan et al., 2013).
Here we find that even for the same aerosol and CCN concentrations, just altering the relationship between aerosol
concentration and ice-nucleating ability can cause changes in daylight TOA outgoing radiation of up to 18.2 W m$^{-2}$ in
our domain.
Our results indicate that the slope of the INP parameterisation with respect to temperature (dlog[INP]/d$T$) is
particularly important: Outgoing total radiation, along with many cloud field and microphysical properties affecting
radiation, were significantly correlated with INP parameterisation slope. Best practise for accurately representing INP
number concentrations based on current knowledge is to utilise parameterisations that link aerosol number and
particle size to INP number concentration (e.g. D10, N12, A13) but that is not enough without also using a
parameterisation in which the temperature dependence of the INP number concentrations matches reality; the
largest differences in domain outgoing radiation existed in this study between simulations using aerosol dependent
parameterisations (D10 and A13). These large variations in outgoing radiation between simulations using different
aerosol dependent INP parameterisations justifies investment in observational campaigns to more effectively
constrain the range of expected INP concentrations and parameterisation slopes in the Saharan dust outflow region,
and other regions dominated by maritime deep convective activity.
The significance of the slope of the INP parameterisation indicates the potential importance of accounting for
differences in aerosol composition in modelling studies. For example, INP derived from marine organics (Wilson et al.,
2015) have a shallower slope than mineral dust INP (Atkinson et al., 2013; Niemand et al., 2012). Furthermore, real-
world INP concentrations are known to have complex temperature dependencies with biological INP, such as soil
borne fungus and plant related bacteria, making significant contributions at the warmest temperatures and mineral
components being more important at lower temperatures (O'Sullivan et al., 2018). The work here suggests that the
presence of biological INP might be to reduce liquid water transport to the upper levels of the cloud, reducing cirrus
anvil extent, but also to increase low cloud fraction. Nevertheless, measurements in the eastern tropical Atlantic
indicate that biological INP in the Saharan dust plumes is at most a minor contribution and that the parameterisations
with shallow slope in Fig. 2 produce too much glaciation at warm temperatures.
The results presented here also present a new framework for understanding the effect of SIP by identifying a potential
relationship between the effect of the Hallett-Mossop process and INP parameterisation slope. The significance of INP
parameterisation slope also highlights the importance of characterising the INP concentration across the entirety of
the mixed-phase temperature range rather than just at one temperature, or in a narrow temperature range, as is
common in many field campaigns. For example, in the ICE-D field campaign, INP concentrations at temperatures
above -7 and below -27°C were not measurable due to experimental and sampling constraints (Price et al., 2018).
Measuring INP over the entire mixed-phase temperature range, throughout which deep convective clouds extend,
conceivably covering around 10 orders of magnitude in INP number concentration, represents a major experimental
challenge. This issue is compounded by the fact that INP spectra cannot reliably be extrapolated to higher or lower
temperatures since our underpinning physical understanding of what makes an effective nucleation site is lacking
(Coluzza et al., 2017; Holden et al., 2019; Kanji et al., 2017).  This work demonstrates the importance of solving these
problems and measuring INP number concentrations across the entirety of the mixed-phase temperature spectrum,
as has been demonstrated in previous work (e.g. Liu et al., 2018; Takeishi and Storelvmo, 2018).
**Data Availability**
The datasets generated and analysed in this study are available from the corresponding author on reasonable request.
**Author Contributions**
REH, AKM, KSC, PRF and BJM contributed to the design, development and direction of the study. REH and AKM set up
and ran the UM-CASIM simulations presented in the paper. REH processed and analysed the UM-CASIM datasets.
JMW, AAH and BJS built and maintained the Met-Office CASIM model used to run the simulations. ZC and RJC
provided processed aircraft data from the ICE-D b933 flight and helped with the comparison of model data with
aircraft measurements. REH, AKM, JMW, AAH, ZC, RJC, KSC, PRF and BJM edited the manuscript.
**Competing interests**
The authors declare no competing interests.
**Acknowledgements**
This work has been funded by European Research Council (ERC, grant 648661 MarineIce) and the Natural Environment
Research Council (NERC, grant NE/M00340X/1). We acknowledge the use of Monsoon, a collaborative High
Performance Computing facility funded by the Met Office and NERC. We acknowledge the use of JASMIN, the UK
collaborative data analysis facility. We obtained moderate resolution imaging spectroradiometer (MODIS) Corrected
Reflectance images from the NASA Worldview website (https://worldview.earthdata.nasa.gov/). Airborne
measurements were obtained from the ICE-D field campaign and specifically the b933 flight on the 21$^{st}$ August 2015.
The ICE-D campaign used the BAe-146-301 Atmospheric Research Aircraft which is operated by Directflight Ltd (now
Airtask) and managed by the Facility for Airborne Atmospheric Measurements (FAAM). At the time of the
measurements FAAM was a joint entity of NERC and the UK Met Office. We thank all the people involved in the ICE-D
campaign.

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

862

863

# Figures

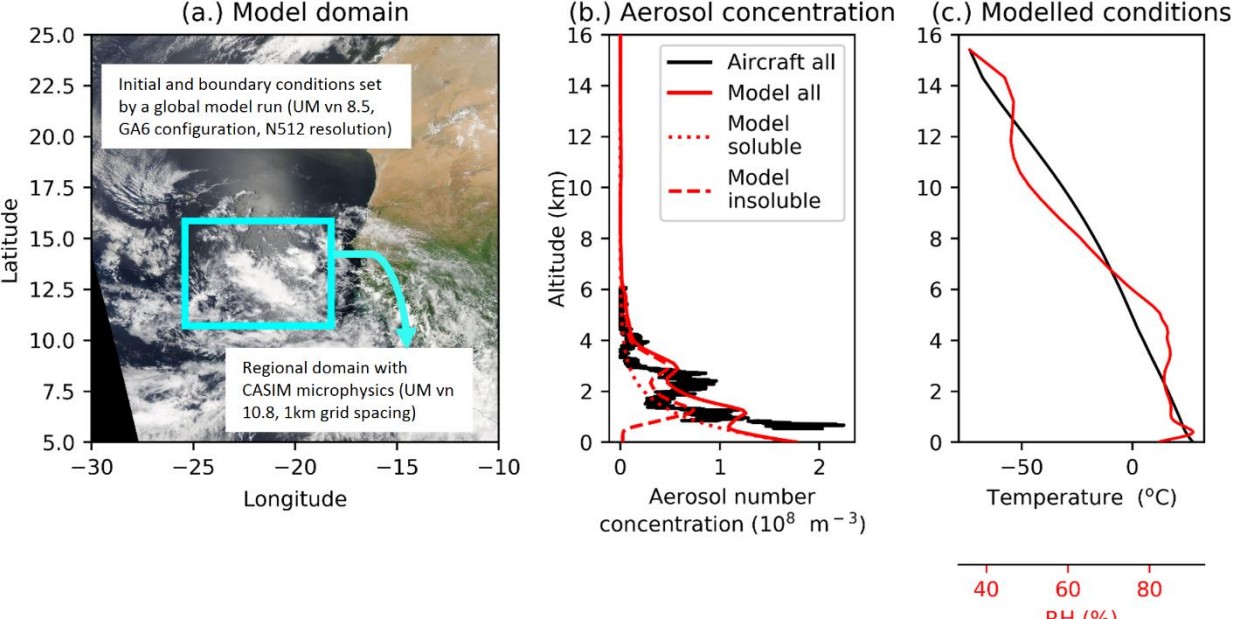

**Figure 1. Modelled domain location and resolution details (a), observed (black line) and modelled (red lines) aerosol concentrations (b), and mean modelled domain mean temperature and relative humidity profiles (c). The observed aerosol profile shown in b was measured using the Passive Cavity Aerosol Spectrometer Probe (PCASP) which captures aerosols between 0.1 and 3μm in size. The insoluble aerosol profile shown in b is extracted from a regional UM vn 10.3 simulation (8 km grid spacing, CLASSIC dust scheme). The modelled aerosol profiles are applied throughout the regional domain shown in a at the start of the simulation (00:00 21st August 2015) and at the boundaries throughout. INP concentrations in the D10, N12 and A13 simulations are linked to the insoluble aerosol profile shown in b. The image shown in (a) are moderate resolution imaging spectroradiometer (MODIS) Corrected Reflectance imagery produced using the MODIS Level 1B data and downloaded from the NASA Worldview website.**

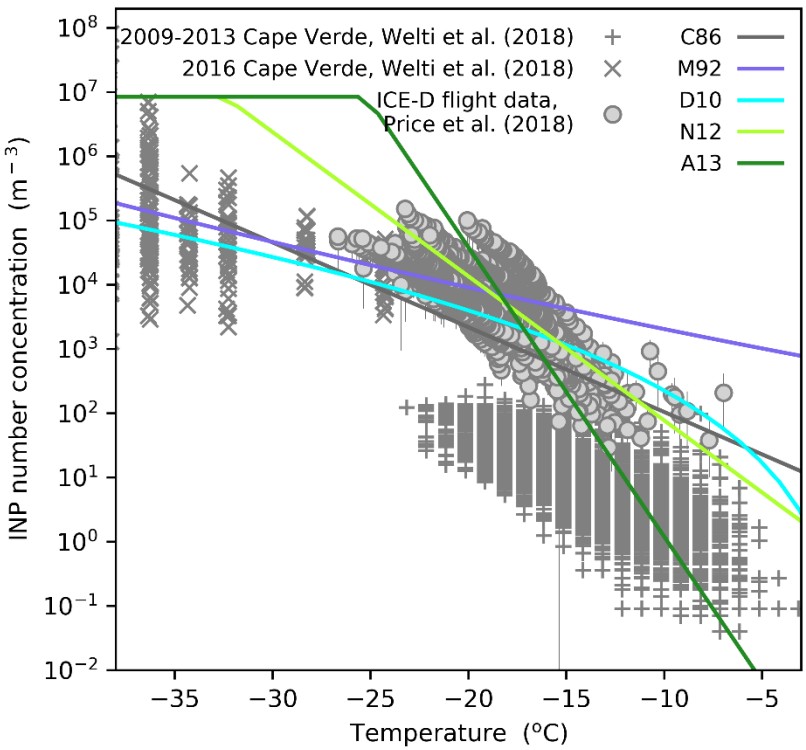

876

**Figure 2. Dependence of INP number concentration on temperature (*d[INP]/dT*) for the five heterogeneous freezing parameterisations simulated in this study (C86, M92, D10, N12, A13) compared to INP number concentrations measured in the eastern Tropical Atlantic(Price et al., 2018; Welti et al., 2018). Parameterisations are shown for the aerosol concentrations at approximately the first freezing level in our simulations (~ 8 cm⁻³). D10, N12 and A13 are dependent on aerosol concentrations, while C86 and M92 are not dependent on aerosol concentration. N12 and A13 are calculated assuming a mean dust particle radius of 0.7 µm. In D10, all particles are assumed to be larger than 0.5 µm. Note that the Welti et al. (2018). Note that the Welti et al. (2018) dataset is from surface INP measurements at Cape Verde while the Price et al. (2018) dataset is measured from an aircraft flown from Cape Verde.**

885

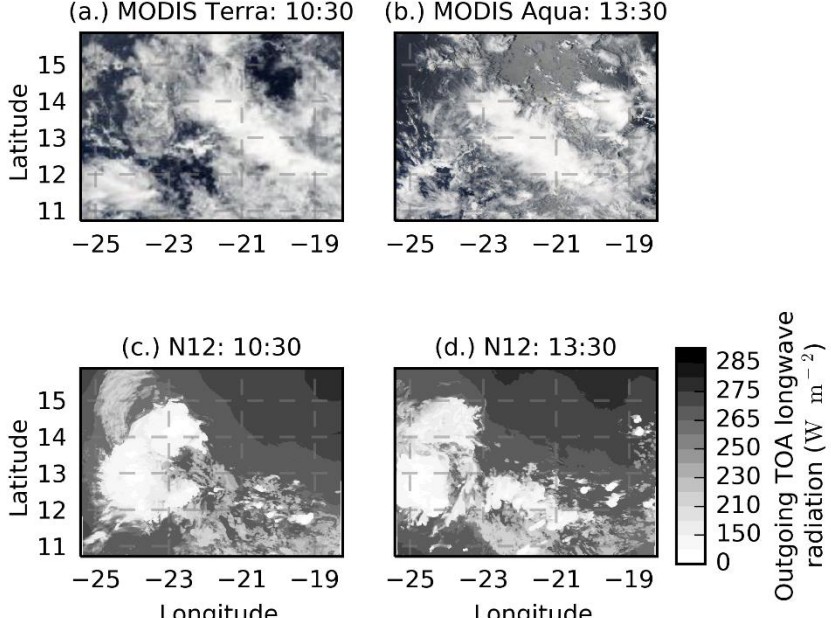

886

**Figure 3. Cloud field evolution. MODIS Terra (a) and Aqua (b) corrected reflectance images of the modelled domain for the 21st of August 2015 and the corresponding simulated top of atmosphere outgoing longwave radiation for the N12 simulation (c, d).Note that the colour bar relates to panels c and d only. Images shown in (a) and (b) are moderate resolution imaging spectroradiometer (MODIS) Corrected Reflectance imagery produced using the MODIS Level 1B data and downloaded from the NASA Worldview website.**


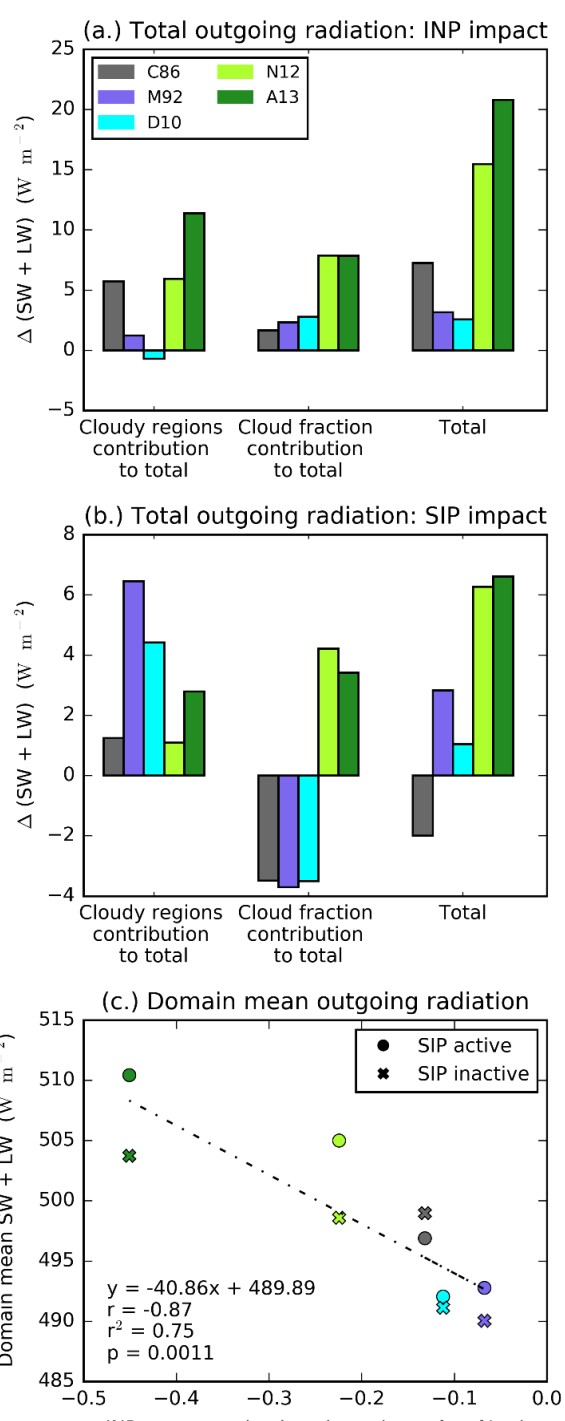


**Figure 4. Effect of INP and secondary production on top of atmosphere (TOA) daytime (10:00-17:00 UTC) outgoing**


**radiation. Effect of INP parameterisation (a) and SIP (a representation of the Hallett-Mossop process) (b) on domain-**


**mean daytime TOA outgoing radiation and total domain-mean daytime TOA outgoing radiation plotted against INP**


**parameterisation slope (c). In (a), the change from the NoINP simulation is shown (INP - NoINP) with SIP active. In (b), the**


**change from SIP_active to SIP_inactive is shown (SIP_active – SIP_inactive). A positive value indicates more outgoing**


**radiation when INP or SIP are active. In (a) and (b), the relative contributions of changes in outgoing radiation from cloudy**


**regions (left) (i.e. $\Delta Rad_{REFL}$ from Eq. (1)) and cloud fraction (middle) (i.e. $\Delta Rad_{CF}$ from Eq. (2)) to the total radiative**


**forcing (right) (i.e. $\Delta Rad_{s-r}$ from Eq. (5) with simulation s referring to simulations with INP active in (a) and to the**


**SIP_active simulations in (b) and simulation r referring to the NoINP simulation in (a) and to the SIP_inactive simulations**


**in (b)) are shown (calculation described in Sect. 2.1.3). In addition to the simulated values, a regression line (n=10) is**
**shown in (c) along with its associated statistical descriptors.**

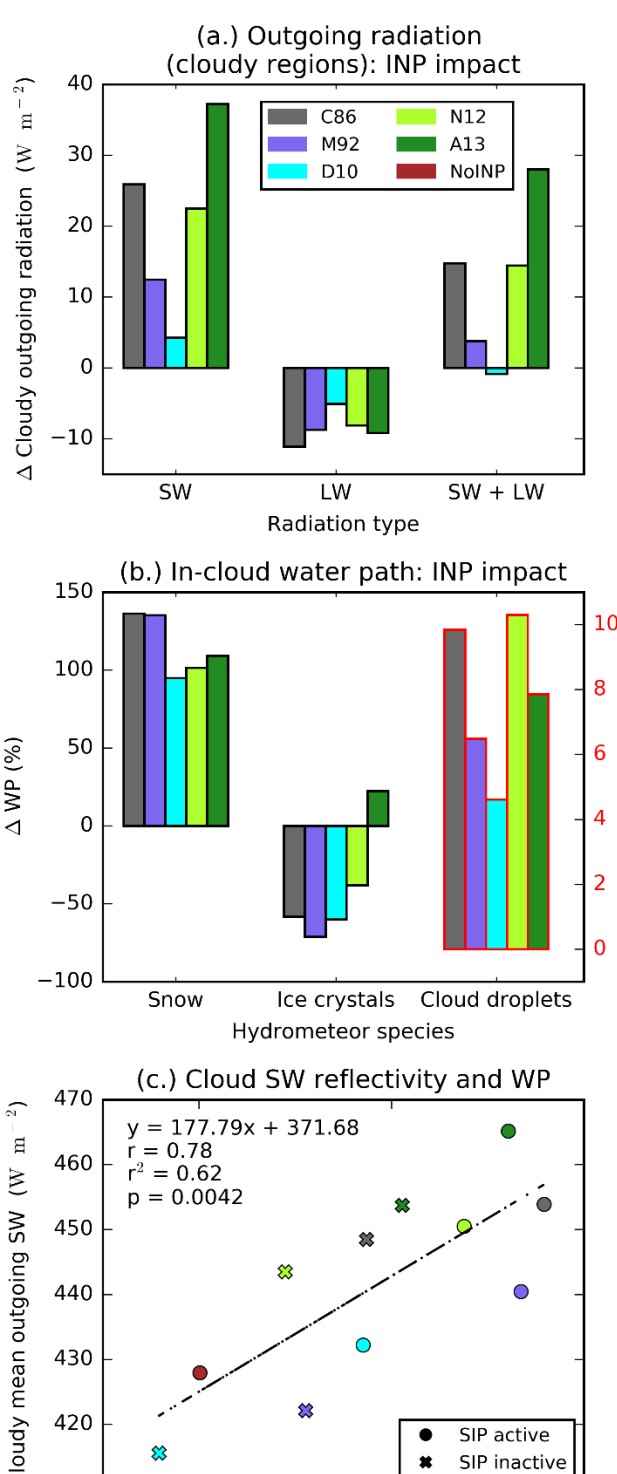


**Figure 5. INP and TOA outgoing daytime (10:00-17:00 UTC) radiation from cloudy regions. Absolute change in outgoing**
**shortwave, longwave and total radiation from cloudy regions relative to the NoINP simulation (i.e. $\Delta Rad_{cl}$ used in Eq.**
**(1)) (a), the percentage change in water path (WP) associated with snow (S), ice crystals (IC) and cloud droplets (CD)**
**relative to the NoINP simulation (b), and mean daytime outgoing shortwave from cloudy regions plotted against the sum**

    **of S, IC and CD water paths (c). Note different scale for CD water path in (b). In addition to the simulated values, a**

    **regression line (n=11) is shown in (c) along with its associated statistical descriptors.**

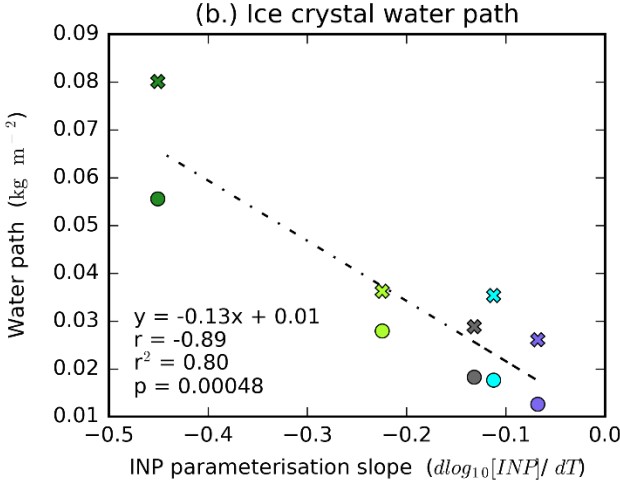

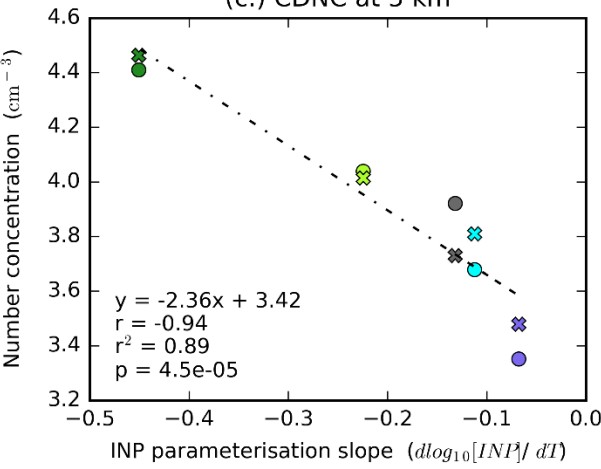


**Figure 6**. **Outgoing daytime (10:00-17:00 UTC) radiation from cloudy regions and INP parameterisation slope. Scatter plots**

**of INP parameterisation slope and total daytime outgoing radiation from cloudy regions (a), in-cloud mean ice crystal**

**(cloud ice only) water path (b), and in-cloud cloud droplet number concentrations at the start of the mixed-phase region**

**(5 km) (c). Also shown are the respective regression lines (n=10) and associated statistical descriptors.**

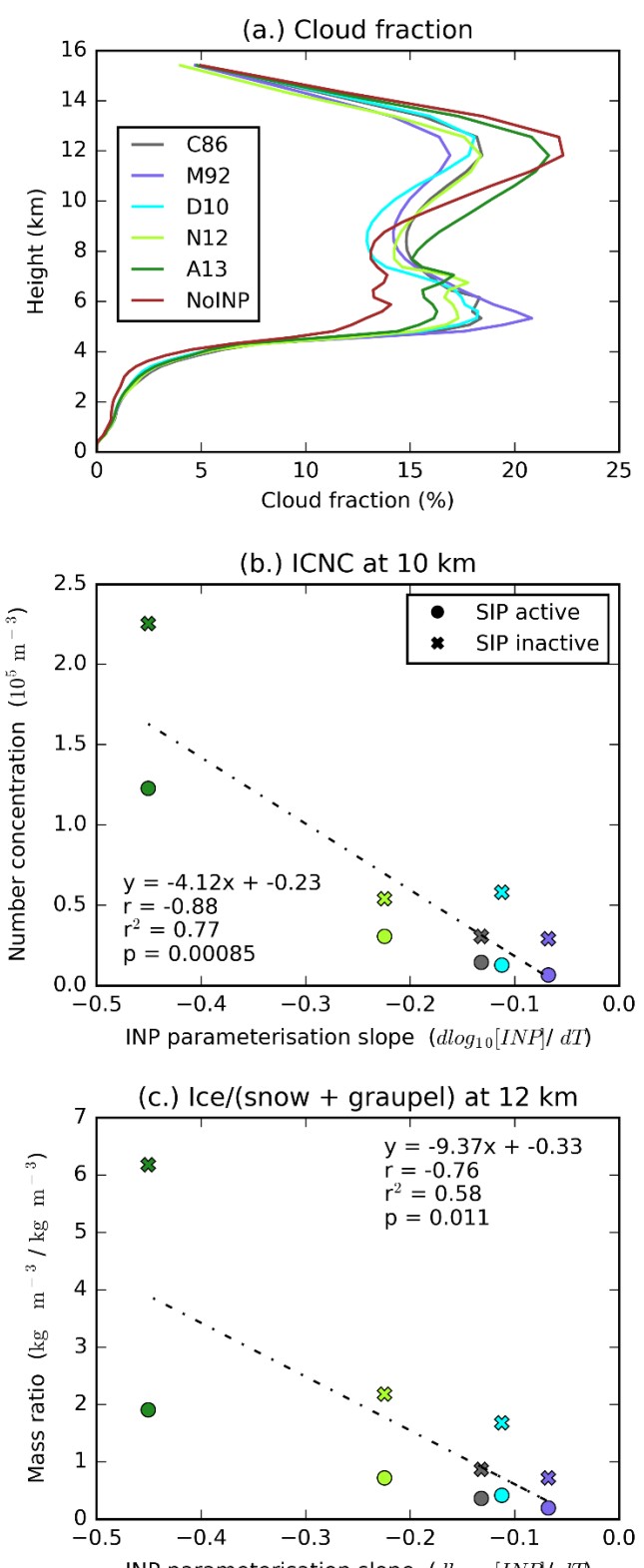

917

**Figure 7. Cloud fraction and INP parameterisation slope. Domain-mean cloud fraction profile (a), INP parameterisation slope plotted against ice crystal number concentration at 10 km (b) and mass ratio of ice crystals to snow plus graupel at 12 km (c). Also shown in (b) and (c) are the respective regression lines (n=10) and associated statistical descriptors.**

921

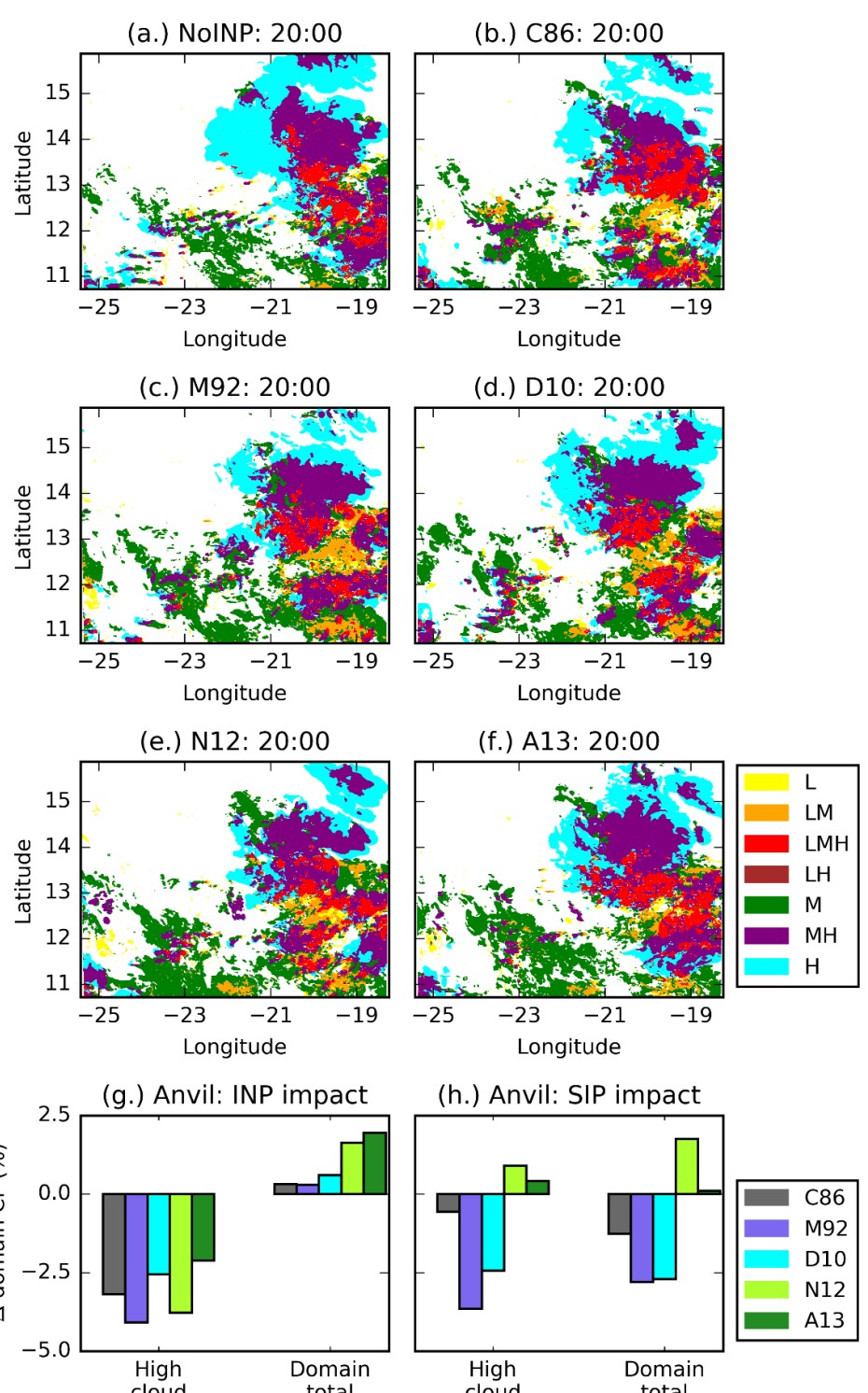

922

**Figure 8. Vertical composition of cloud. 2D distribution of cloud type at 20:00 for all six SIP_active simulations (a-f), as well as anvil and domain cloud fraction change (10:00-24:00 UTC) due to INP (g) and due to SIP (h). Clouds are categorised according to their altitude into low (L, <4 km), mid (M, 4-9 km) and high (H, >9 km) levels and mixed category columns if cloud (containing more than $10^{-5}$ kg kg$^{-1}$ condensed water from cloud droplets, ice crystals, snow and graupel) was present in more than one of these levels (a more detailed description can be found in Sect. 2.1.4). A positive value in (g) or (h) indicates higher values when INP (g) or SIP (h) are active.**



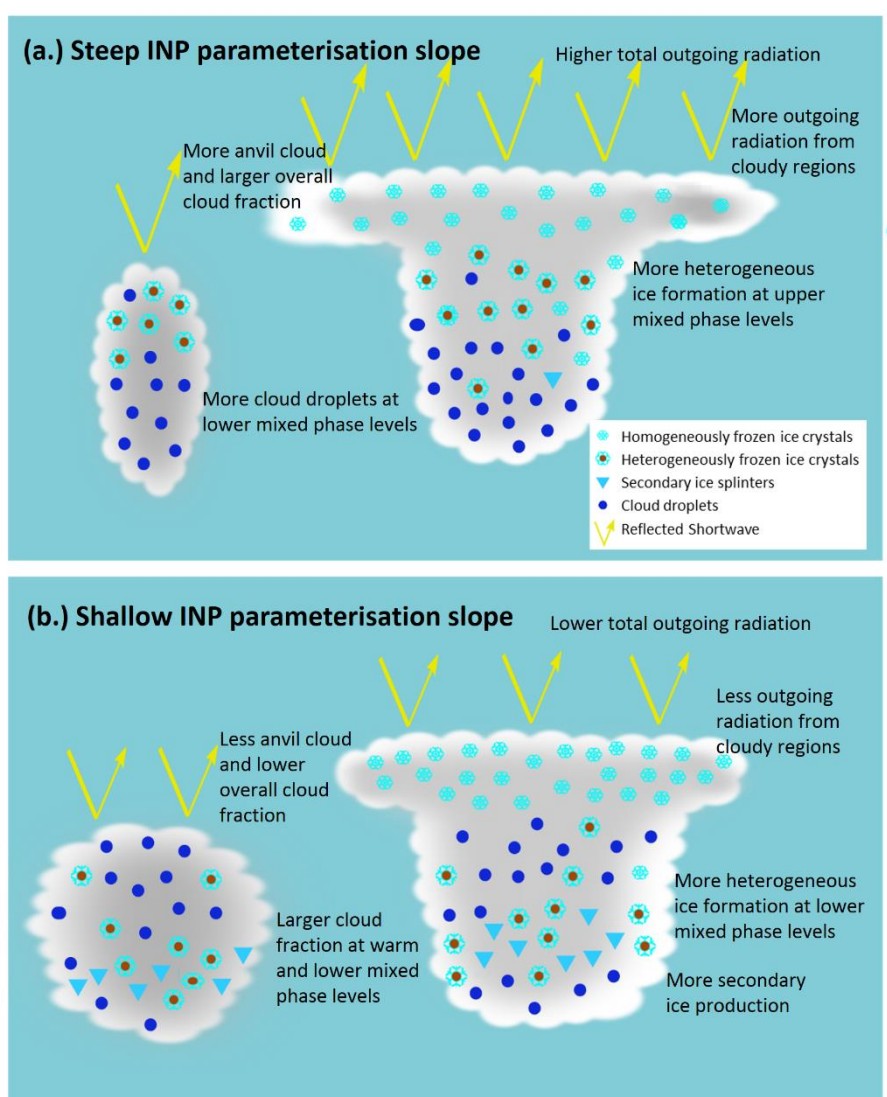


**Figure 9. Schematic of the main effects of INP parameterisation slope (i.e. a steep (a) or shallow (b) temperature dependence of INP number concentrations).**


# Appendix A

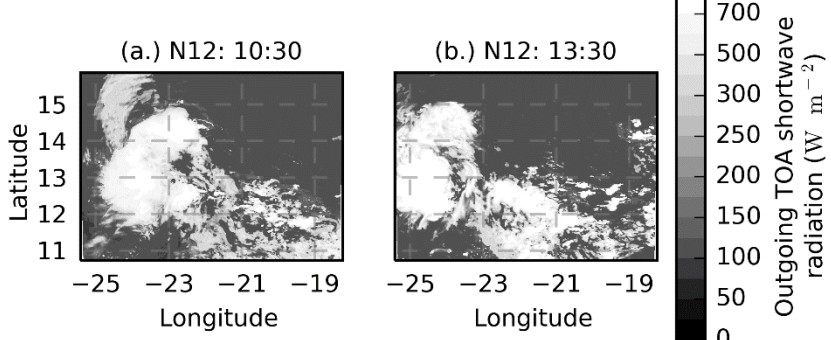

**Figure A1. The cloud field. Simulated top of atmosphere outgoing shortwave radiation for the N12 simulation at 10:30 (a) and 13:30 (b).**

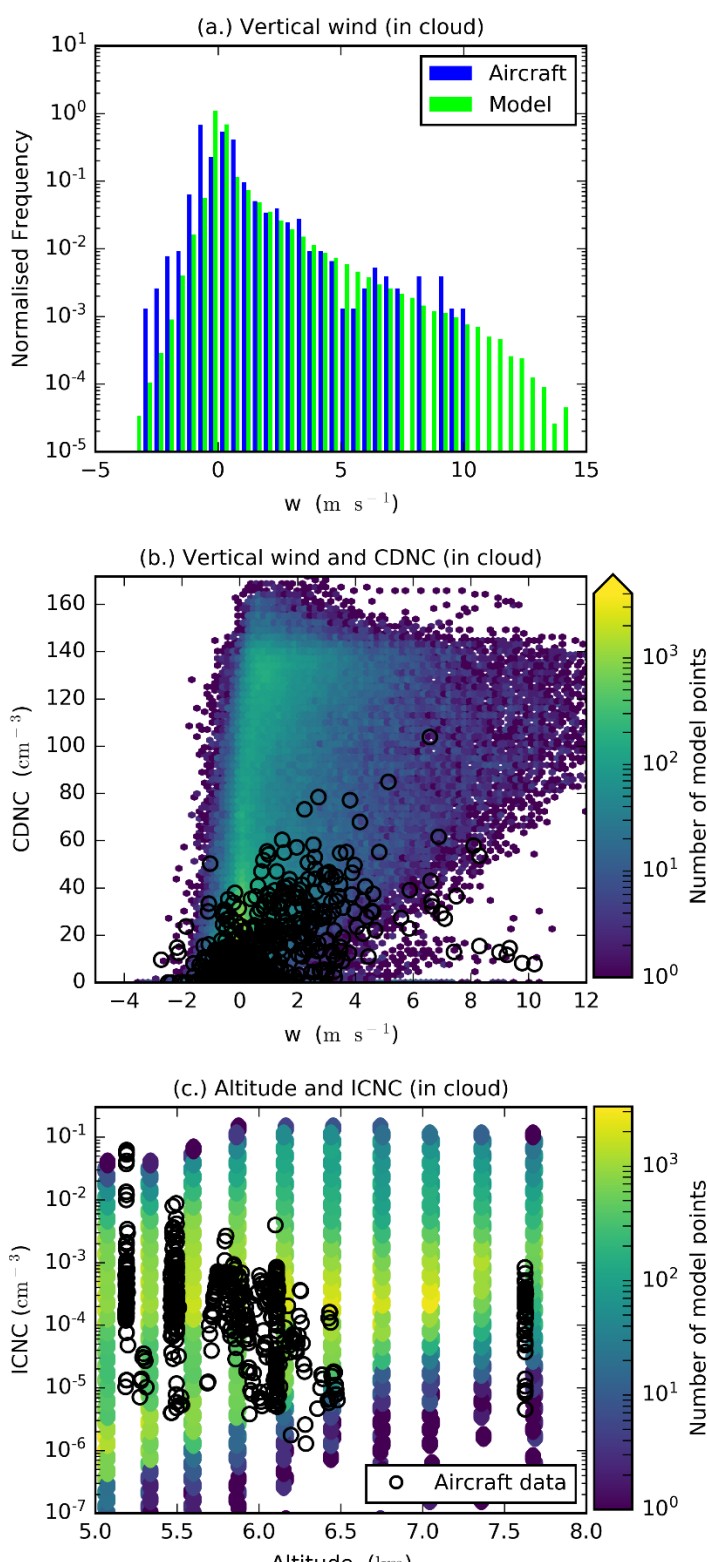

940

**Figure A2. Comparison of observed conditions from the b933 ICED field campaign flight on the 21st August 2015 and the modelled conditions. Vertical wind speed from the model and aircraft data (a), a 2D histogram of modelled vertical wind against cloud droplet number concentration (CDNC) (b) and altitude plotted against ice crystal number concentration (ICNC) (c) with the aircraft data overlaid. Modelled values are selected from clouds between 10 and 150 km$^2$ in size from the N12 simulation.**

946

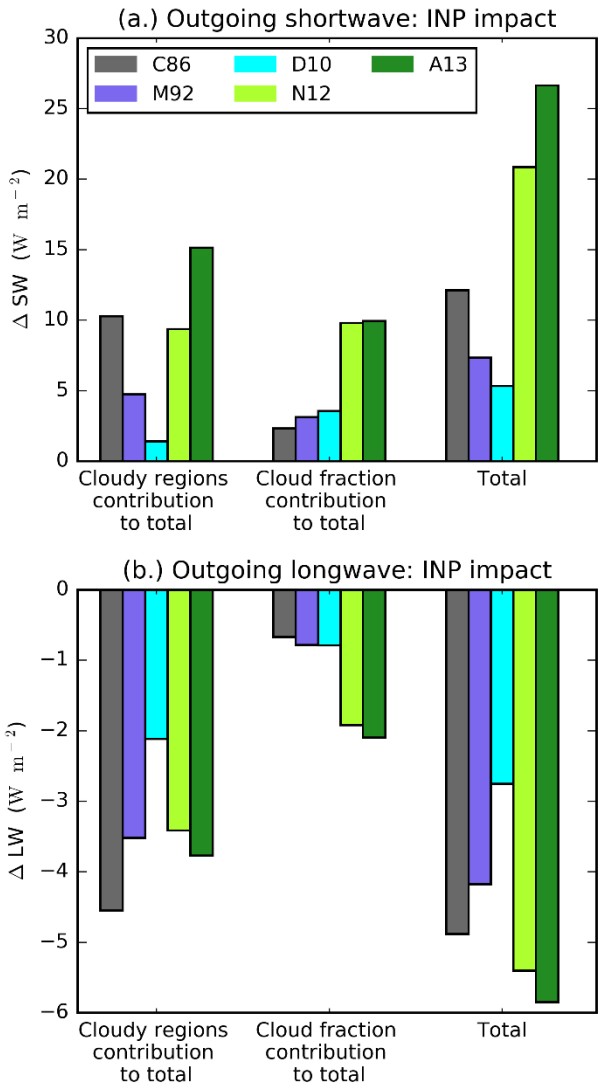

947

**Figure A3. Effect of INP on domain-mean TOA outgoing daytime (10:00-17:00 UTC) shortwave and longwave radiation.**
**The change from the NoINP simulation is shown (INP - NoINP). A positive value indicates more outgoing radiation when**
**INP are present. The contributions of changes in outgoing radiation from cloudy regions (left)** *(i.e. $\Delta Rad_{REFL}$ from Eq. (1))*
**and cloud fraction (middle)** *(i.e. $\Delta Rad_{CF}$ from Eq. (2))* **to the total radiative forcing (right)** *(i.e. $\Delta Rad_{s-r}$ from Eq. (5)*
**with simulation s referring to the simulations with INP active and simulation r referring to the NoINP simulation) are also**
**shown (calculation described in Sect. 2.1.3).**


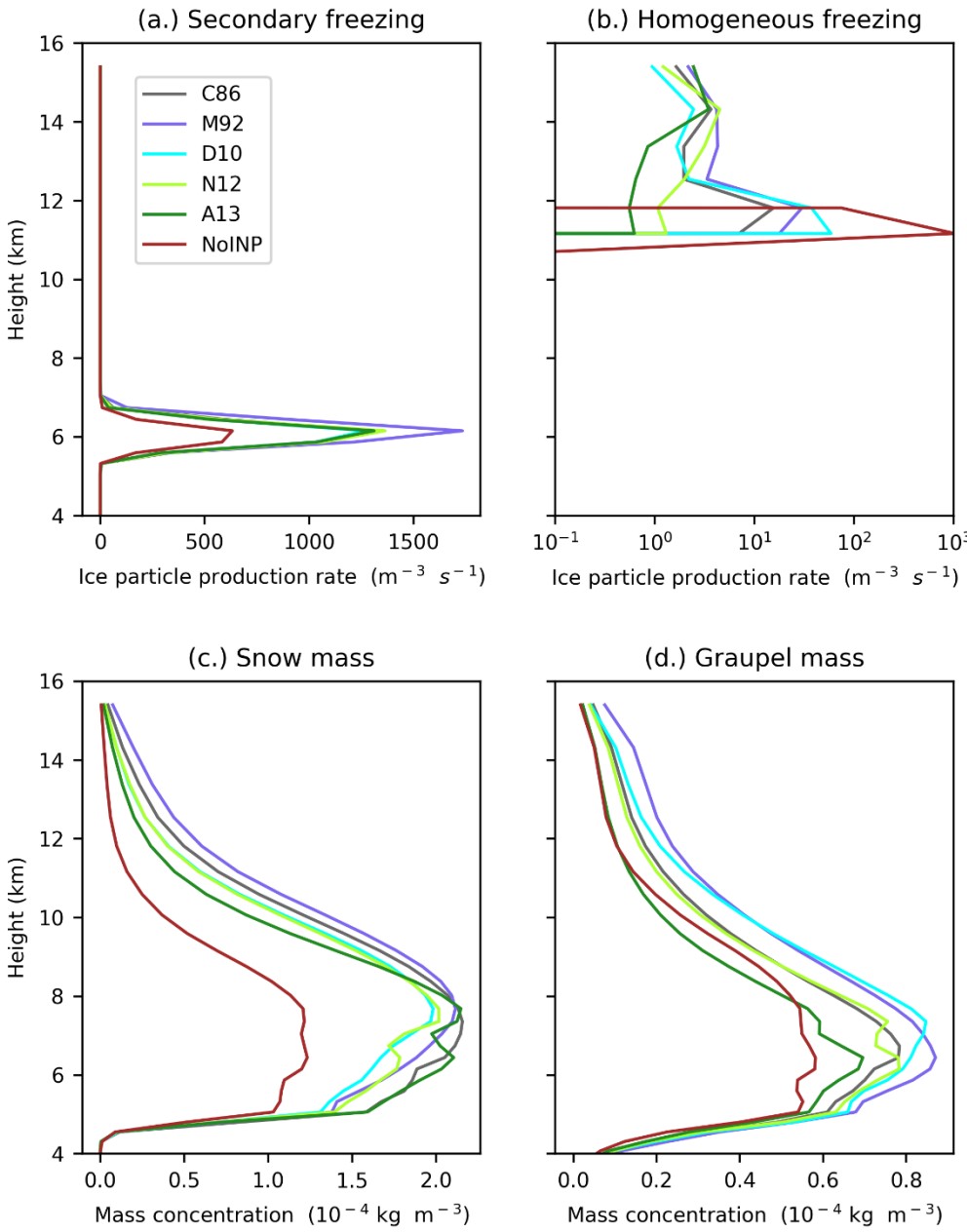


**Figure A4. Profiles of some microphysical properties of the simulated clouds. Mean in-cloud ice particle production rates from secondary (b) and homogeneous (c) freezing, snow mass concentration (c) and graupel mass concentration (d).**



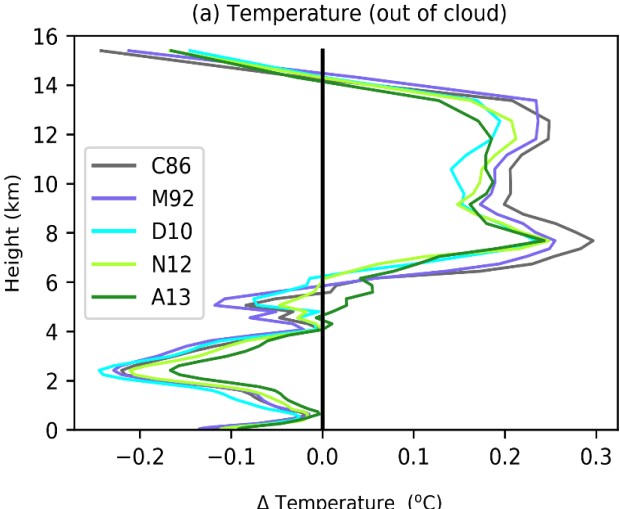

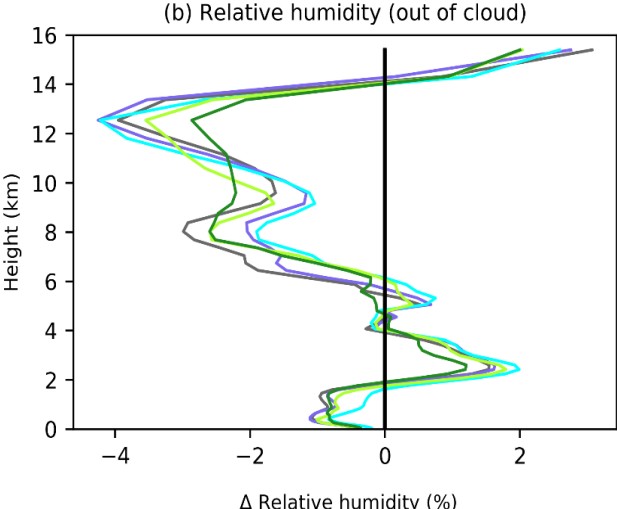

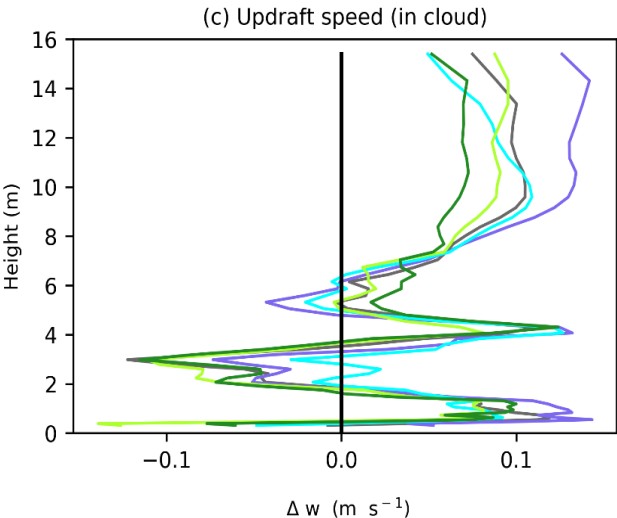


**Figure A5. Effect of INP on domain mean out of cloud temperature (a) and relative humidity (b), and in cloud updraft speed (c). The difference from the NoINP simulation is shown, a positive value indicates a higher value when INP is present.**