# Peer review of "The temperature dependence of ice-nucleating particle"

_Atmospheric Chemistry and Physics, 2020_

## Referee Comment (RC1) · Anonymous Referee #1 · 17 Aug 2020

**Review of Hawker et al. (2020): The nature of ice-nucleating particles affects the radiative properties of tropical convective cloud systems, submitted to Atmospheric Chemistry and Physics.**

**General comments:**

The authors present a very interesting study on the effect of different INP parametrizations on the radiative budget in Tropical Atlantic deep convective cloud fields. In particular, they show that INP parametrizations with a larger temperature-dependency lead to a larger increase in domain-mean daytime top of atmosphere outgoing radiation. In contrast to previous work by other authors, they present data that indicates primary ice production to be relevant also in presence of secondary ice formation. It should be clarified that the INP parameterizations tested against the Hallet-Mossop process explicitly, since this is the only SIP considered in the model.

The writing (from an editorial standpoint) is to be commended. The methodology appears stringent and valid with one minor aspect to be clarified. The work addresses relevant scientific atmospheric questions with impacts on global climate simulations. The title speaks of the nature of INPs, but the paper never refers back what "the nature" explicitly is. The topic of the paper is well suited for ACP. I recommend the manuscript for publication if the above and following comments below are addressed:

**Specific comments:**

Line 19-20: The paper presented tests the effect of 5 parameterizations on radiation and SIPs but the word parameterization doesn't shown up even once in the abstract. Also, the work is not explicitly testing the impact of the nature of the INPs (i.e. bio and physico-chemical properties of INPs). It is testing the impact of the various parameterizations which are based on desert dust, feldspar and continental aerosol, so there isn't an explicit testing of physico-chemical properties of INP on the radiation. The C86 and M92 are not aerosol based either. I suggest being more explicit about what this work does. Same goes with the title, the nature of the INP is not tested as much as the type of parameterization.

Line 22: should add "due to the Hallet-Mossop process" after "....secondary ice production" since the paper tests the parameterizations against one secondary ice process, not all.

Line 51: Add Peckhaus et al. (2016) to the references who also studied different types of feldspars and reported  $n_s$ .

line 53: Are e.g. mineral dust events not driven by meteorological factors (gust fronts, lack of precipitation, convective instability, trade winds, etc.) and have a major impact on INP concentrations?

Line 59-60: Not sure what the authors means here. Can this sentence be elaborated upon, or restructured?

Page 6 (line 165). Maybe it would help the reader to also convert the smallest allowable size of ice to an idealized diameter of a sphere, e.g. (i.e.  $10^{-18}$  kg or ~0.1 µm in diameter)

Line 183: What errors would you expect from SOCRATES not responding to changes in ice crystal or snow number concentration, or any changes to rain or graupel species? An over- or underestimation in radiative processes? Also, how does the model then account for changes in in ICNC due to heterogeneous freezing?

Line 189: If LW radiation is only calculated for daytime, how biased is this estimate, since LW would be most effective (trapping outgoing radiation) at night time, so how will this bias the results presented for outgoing LW radiation.

Line 196: should "( $\Delta Rad_{REFL}$ )" come after the word "...difference" in the sentence? It seems to appear too early in the sentence.

Line 197: What do s and r stand for? It is hard to follow the equations below without knowing some sort of physical definition of s and r.

Line 201: I am not sure if I followed equation 2 correctly, but could  $\Delta cf$  in this equation be replaced with (Rads,cl – Rads,cs) since you are looking at the contributions of cloud fraction to radiation between simulation s and r? If this is correctly understood, I would replace the  $\Delta cf$  with (Rads,cl – Rads,cs), to make it easier to follow equation 2. Or define  $\Delta cf$  more explicitly than has been done in line 203.

Line 239: If the evolution of the clouds are not being discussed further, then no need to show the plots in figure 2e, f,g and h. The most useful plots are a-d since they compare the satellite with the model. Since there are no comparisons to the satellite for the other times, it doesn't give much of a validation. Also to show that the model produces a complex realistic cloud field can be demonstrated with Fig2c and 2d, so that c-h are not necessary.

Line 242: Same comment as above with Fig A2, panel c, d, e and f are not necessary.

Line 255: "flow" should be "flown"

Line 272: In the noINP case, can ice crystals that formed via homogeneous freezing, fall to lower levels and initiate secondary ice processes?

Lines 281 to 283: similar to comment above, the comparison of noINP to simulations with INP parameterization demonstrates an enhancement in outgoing radiation for D10 and A13. Can the authors clarify here that noINP simulation excludes any contribution of SIPs that could result from ice settling from higher altitudes to warmer regions of the clouds? Or is such a contribution included in this assessment?

To me this seems possible to diagnose in the model, if the assessment of precipitation evaporating results in higher humidity such that the LWP due to increase cloud droplet number. Then why isn't the possibility of ice crystals or snow settling through the clouds at Hallet-Mossop relevant temperatures allowed to produce secondary ice?

Line 285-286: "Radiative changes from the NoINP simulation to the inclusion of INP are caused mainly by .."

It is not cleat to me what is meant by this statement. I think it means the difference in radiation between the NoINP and the INP simulations mostly arises from the changes in outgoing SW radiation, but if that is the case why not state it more simply, i.e. the difference in radiation between NoINP and INP simulations ...

Lines 291 – 292: The reported comparatively small change in TOA radiation when SIP is active relative to when it is inactive mentioned here, could this be because in the homogeneous freezing run SIP is by default assumed to be absent (i.e. settling of crystals to warmer Hallet-Mossop regions for secondary ice is not represented in the model)?

Line 293: The relationship between the slope of the parameterization and the outgoing radiation is implicit and not explicit. So stating that it is the key determinant seems very strong here, without clarifying that the slope is representative of the INP concentration as a function of T (Fig 1). It is the T at which a certain proportion of INPs are active is they key determinant and the physical reason for the strong influence on available supercooled liquid to be transported to higher altitudes of the MPC region. Could this relationship be clarified to invoke the temperature dependence rather than just stating the slope of the parameterization is the key determinant?

Line 303-304: This sounds counterintuitive to me because compared to the NoINP simulation, the increase in INP should increase the ICNC and decrease the cloud lifetime or outgoing SW (reflectivity) compared to otherwise what would be a more reflective liquid cloud with less ice. However further down the authors do explain why they observe this, because the liquid water path increases in the warmer part of the cloud which increases the outgoing SW. However, I find this assessment to be biased, without accounting for potential SIP in the NoINP simulation due to settling ice crystals.

Line 309/Figure 4b: I would have expected the water path due to snow to decrease because snow should leave the cloud faster than ice crystals? So why is the water path due to ice crystals decreasing? Or does it have to do with the categorisation of when a hydrometeor is considered a snow flake vs. an ice crystal in the model? Also, I would have expected that the increase in water path should be the lowest for the M92 and not for the D10, since the M92 has the shallowest slope?

Lines 313-320: If the precipitation is increased, then the snow and graupel should not be part of the cloud anymore and thus not contribute to the increased reflected shortwave radiation. If the increased condensate is falling as precipitation such that it is resulting in an increased humidity thus increasing the LWP from an increase in cloud droplets, how can it also contribute to increasing the outgoing shortwave in the cloud? It should either be classified as increased precipitation below cloud or increased condensate in-cloud leading to outgoing shortwave.

Lines 321 – 334: This explanation makes more sense and sounds stronger and more convincing to me than the explanation in lines 308 to 320. Perhaps it would be better to shift the order of these paragraphs and explain the higher outgoing shortwave by the increased CDNC at lower altitudes due to increased LWP from lower freezing rates arising from steeper INP parameterizations (which imply very little het. freezing at small supercooling).

Line 331: clarify statement more, I suggest (italics is suggested part) something like "...due to lower rates of heterogeneous freezing *at the mid-bottom region of the mixed-phase cloud (lower supercooling, Fig. 1)* and SIP at ..."

Line 339: clarify by inserting "cloud fraction due to" i.e. sentence should read

"...offset somewhat by decreases in the cloud fraction due to homogeneous freezing in the  $\sim 10 - 14$  km regime (Fig 6a)"

Line 358-360: If this is true, (and it sounds like a good explanation), shouldn't the decrease in outgoing LW shown in Figure 4a be the highest for A13 and not for C86. Because A13 results in the highest number of ICNC at the top of the MPC region and in the homogeneous freezing region therefore should trap most of the outgoing LW radiation thus giving the most decrease in the outgoing LW.

Line 379:. Change to "It has been argued that the observed (or derived) primary ice particle production rate...". Otherwise, the statement is false, because if the primary production rate is high, the secondary ice production (H-M process) would be low but still present, primary ice production would in fact dominate cloud properties.

Line 384/ line 219/lines 440-445: Most relevant comment. You show that higher primary ice production rates in the temperature range between 253 and 238 K, e.g. in A13, have a large impact on the total on top of atmosphere outgoing radiation, yet you exclude SIP which are active at colder temperatures than 265 K. Can you elaborate on the expected impact/uncertainty in your results and concluding statement, (SIP is less important than primary ice production) stemming from your simplification that SIP is only including Hallett Mossop process? Please justify why your concluding statement is valid.

Line 383-384: This is an important outcome of the study, but should be caveated with the notion that other possible known and unknown SIPs are not considered. The authors in part do that by acknowledging in parenthesis that the SIP considered is the H-M, but I think they could go one step further in saying that this could change with more parameterizations and quantification becoming available for lower temperatures where SIP becomes important say below 265 K (Lauber et al., 2018).

Page 14 (line 385). An average impact comparison in the text might be supportive for the reader (e.g. mean over all parameterizations total INP impact 9.8 W/m2 to total SIP impact 2.7 W/m2)

Line 453-455: A possible explanation for this statement should be given here in in the conclusions, since this is an important point or outcome of the study. Have the other studies that are mentioned in these lines also only tested the influence of the Hallet-Mossop process? If not, this should be clarified. Further, since this has evaluated the influence of SIP due to the HM process, it should be stated here in the conclusions.

So this conclusion is true, when the SIP being considered is HM. But it remains open if the conclusions would still hold if freeze shattering and other mechanisms (e.g., as described in Lauber et al., 2018) were included in the models.

Line 496: In addition to Holden et al. (2019) the authors could add Coluzza et al. (2017) and Kanji et al. (2017) since that lack of knowledge on what constitutes the identity of an active site was already discussed in these publications.

Line 497: The last statement here has surely been mentioned before by other authors in numerous publications. While it is valuable that the authors also come to this conclusion (need for INP measurement across the entirety of the MPC regime), this study is not the first to recommend such an outlook and the sentence can be modified to say "..as reported before by XX.."

Figure 2: Should there not be a "radiation" in the color bar caption, e.g. Outgoing OA longwave radiation  $(W m^{-2})$ ?

Figure 5: Colour legend is missing. I suggest adding it here even if it is the same as previous figures for ease of reading.

Appendix A, page 34 (Figure A1). The three kind of blue lines are not easy to distinguish from the black line.

Appendix A, page 34 (Figure A1). What does the unit of  $/ 10^8 \text{ m}^{-3}$  mean? Is it two particles per  $10^8 \text{ m}^{-3}$  of volume?

Figure A3 panel a: what are the regions filled with black colour? Could the colors be changed so that the contrast between green and blue is better visible? If black is just the border of the bars, I suggest removing the borders since it reduces clarity of the plot suggesting that there is a third colour.

Line 810: Histograms should be singular not plural.

**References**

Coluzza, I., Creamean, J., Rossi, J. M., Wex, H., Alpert, A. P., Bianco, V., Boose, Y., Dellago, C., Felgitsch, L., Fröhlich-Nowoisky, J., Herrmann, H., Jungblut, S., Kanji, A. Z., Menzl, G., Moffett, B., Moritz, C., Mutzel, A., Pöschl, U., Schauperl, M., Scheel, J., Stopelli, E., Stratmann, F., Grothe, H., and Schmale, G. D.: Perspectives on the Future of Ice Nucleation Research: Research Needs and Unanswered Questions Identified from Two International Workshops, 8, doi:10.3390/atmos8080138, 2017.

Holden, M. A., Whale, T. F., Tarn, M. D., O'Sullivan, D., Walshaw, R. D., Murray, B. J., Meldrum, F. C., and Christenson, H. K.: High-speed imaging of ice nucleation in water proves the existence of active sites, 5, eaav4316, doi:10.1126/sciadv.aav4316, 2019.

Kanji, Z. A., Ladino, L. A., Wex, H., Boose, Y., Burkert-Kohn, M., Cziczo, D. J., and Krämer, M.: Overview of Ice Nucleating Particles, in: Ice Formation and Evolution in Clouds and Precipitation: Measurement and Modeling Challenges, 2017.

Lauber, A., Kiselev, A., Pander, T., Handmann, P., and Leisner, T.: Secondary Ice Formation during Freezing of Levitated Droplets, 75, 2815-2826, doi:10.1175/JAS-D-18-0052.1, 2018.

Peckhaus, A., Kiselev, A., Hiron, T., Ebert, M., and Leisner, T.: A comparative study of K-rich and Na/Ca-rich feldspar ice-nucleating particles in a nanoliter droplet freezing assay, Atmos. Chem. Phys., 16, 11477-11496, doi:10.5194/acp-16-11477-2016, 2016.

---

## Referee Comment (RC2) · Anonymous Referee #2 · 6 Oct 2020

See attachment.

Please also note the supplement to this comment:
https://acp.copernicus.org/preprints/acp-2020-571/acp-2020-571-RC2-supplement.pdf

––––––––––––––––––––––––

---

## Author Comment (AC1) · 17 Nov 2020

**Replies to review RC1**

Review of Hawker et al. (2020): The nature of ice-nucleating particles affects the radiative properties of tropical convective cloud systems, submitted to Atmospheric Chemistry and Physics.

Thank you for your helpful comments on our paper. Note that as a result of additional figures in response to review RC2, the figure numbers in our responses to this review are different to those in the original manuscript.

General comments:

The authors present a very interesting study on the effect of different INP parametrizations on the radiative budget in Tropical Atlantic deep convective cloud fields. In particular, they show that INP parametrizations with a larger temperature-dependency lead to a larger increase in domain-mean daytime top of atmosphere outgoing radiation. In contrast to previous work by other authors, they present data that indicates primary ice production to be relevant also in presence of secondary ice formation. It should be clarified that the INP parameterizations tested against the Hallet-Mossop process explicitly, since this is the only SIP considered in the model.

Change to paper: We have made clearer in the text that the only SIP tested in this study was the Hallett Mossop process. We have also highlighted more explicitly that the results are only relevant to the Hallett Mossop process and future studies should explore the impact of INP in the presence of other secondary ice production mechanisms. We have detailed our changes below, but for example in the abstract, we now state '*The controlling effect of the INP temperature dependence is substantial even in the presence of Hallett Mossop secondary ice production…..*'

The writing (from an editorial standpoint) is to be commended. The methodology appears stringent and valid with one minor aspect to be clarified. The work addresses relevant scientific atmospheric questions with impacts on global climate simulations. The title speaks of the nature of INPs, but the paper never refers back what "the nature" explicitly is. The topic of the paper is well suited for ACP. I recommend the manuscript for publication if the above and following comments below are addressed:

Change to paper: We have rephrased the title and any mention of the 'nature of INP' in the text to refer instead to the INP parameterisation choice or the temperature dependence of the INP number concentration.

Specific comments:

Line 19-20: The paper presented tests the effect of 5 parameterizations on radiation and SIPs but the word parameterization doesn't shown up even once in the abstract. Also, the work is not explicitly testing the impact of the nature of the INPs (i.e. bio and physico-chemical properties of INPs). It is testing the impact of the various parameterizations which are based on desert dust, feldspar and continental aerosol, so there isn't an explicit testing of physico-chemical properties of INP on the radiation. The C86 and M92 are not aerosol based either. I suggest being more explicit about what this work does. Same goes with the title, the nature of the INP is not tested as much as the type of parameterization.

Change to paper: The title has been changed to "The **temperature dependence** of ice-nucleating particle **concentrations** affects the radiative properties of tropical convective cloud systems" and the lines in question have been changed to "Our results show that the domain-mean daylight outgoing radiation varies by up to 18 W m$^{-2}$ depending on **the chosen INP parameterisation**. The key distinction between different INP **parameterisations** is the temperature dependence of ice formation, which alters the vertical distribution of cloud microphysical processes."

Line 22: should add "due to the Hallet-Mossop process" after "....secondary ice production" since the paper tests the parameterizations against one secondary ice process, not all.

Change to paper: We have added 'Hallett-Mossop': 'The controlling effect of the INP temperature dependence is substantial even in the presence of Hallett Mossop secondary ice production'

Line 51: Add Peckhaus et al. (2016) to the references who also studied different types of feldspars and reported ns.

Change to paper: As suggested.

line 53: Are e.g. mineral dust events not driven by meteorological factors (gust fronts, lack of precipitation, convective instability, trade winds, etc.) and have a major impact on INP concentrations?

Reply: The reviewer is correct that meteorological factors strongly influence dust concentrations in the Cape Verde region, but a number of studies have found that simple meteorological variables such as temperature and pressure are poor predictors of INP number concentrations.

Change to paper: We have edited this to read 'Although INP concentrations do not simply correlate with meteorological variables such as pressure and temperature…'. We have also added (Price et al., 2018) to the citation which found large variability in INP number concentrations of 'African continental 'INP filter samples during the ICE-D field campaign (see Figure 9 and the associated discussion in Section 3.1 of Price et al. (2018)).

Line 59-60: Not sure what the authors means here. Can this sentence be elaborated upon, or restructured?

Reply: The sentence in question highlighted that the studies in question did not alter the temperature dependence of the INP number concentration or used the same INP parameterisation in their approach.

Change to paper: The sentence has been changed to "However, in these model studies perturbations to INP number concentrations have predominantly involved uniform increases in aerosol or INP concentrations with all simulations using the same INP parameterisation (Carrió et al., 2007; Connolly et al., 2006; Deng et al., 2018; Ekman et al., 2007; Fan et al., 2010; Gibbons et al., 2018; van den Heever et al., 2006; Phillips et al., 2005)**, i.e. the temperature dependence of INP number concentrations was not altered.**"

Page 6 (line 165). Maybe it would help the reader to also convert the smallest allowable size of ice to an idealized diameter of a sphere, e.g. (i.e. 10-18 kg or ~0.1 μm in diameter)

Change to paper: As suggested, we have added the approximate effective radius to the text: "The ice splinters produced by the representation of the Hallett Mossop process are the smallest allowable size of ice in the model (i.e. $10^{-18}$ kg**, volume radius ~0.11 μm**)"

Line 183: What errors would you expect from SOCRATES not responding to changes in ice crystal or snow number concentration, or any changes to rain or graupel species? An over- or underestimation in radiative processes? Also, how does the model then account for changes in in ICNC due to heterogeneous freezing?

Reply: As the reviewer notes, the lack of consideration of ice and snow particle number by the SOCRATES radiation scheme is an important limitation of the results presented here. As such changes to ICNC due to heterogeneous freezing are not represented in the calculations, except through any corresponding changes in ice crystal water path. Due to the complex interactions between multiple hydrometeor types in these simulations, we don't feel able to give an accurate estimate of whether our results would under- or overestimate the radiative process. We acknowledge that the outgoing radiation simulated here may change with the inclusion of particle number and size from snow or ice, or from the inclusion of rain and graupel, and this is

clearly stated and discussed in Section 4. The difference the inclusion of particle number from snow or ice would make to the outgoing radiation will vary across the domain and over time depending on whether the 'real' modelled effective radius was larger or smaller than the one implicitly assumed by the radiation code.

Line 189: If LW radiation is only calculated for daytime, how biased is this estimate, since LW would be most effective (trapping outgoing radiation) at night time, so how will this bias the results presented for outgoing LW radiation.

Reply: We decided to only calculate the radiative differences for the daytime hours because the simulation is 24 hours in length and includes 6 hours of spin up. Including the night-time hours in the radiation calculations would either mean including the spin up or averaging over 18 hours which gives an artificial dependence on simulation length (because we would be excluding some of the dark hours). Therefore we felt the most consistent and meaningful way to report the radiation differences was to analyze the daytime hours only. The inclusion of night-time values would reduce the contribution of shortwave to the total outgoing radiation differences shown in Figure 4a. The difference between including and excluding night-time values when averaging the outgoing longwave radiation was checked and found to be negligible.

Line 196: should "($\Delta Rad_{REFL}$)" come after the word "…difference" in the sentence? It seems to appear too early in the sentence.

Change to paper: The sentence has been changed as follows to make the definition of ($\Delta Rad_{REFL}$ clearer:

"The cloudy regions contribution, **i.e. the difference in outgoing radiation between two cloudy regions due to changes in cloud albedo or thickness ignoring any changes in cloud fraction,** $(\Delta Rad_{REFL})$ to a domain radiative difference…."

Line 197: What do s and r stand for? It is hard to follow the equations below without knowing some sort of physical definition of s and r.

Reply: s refers to a generic sensitivity simulation (e.g. a simulation using a specific INP parameterisation while r refers to a generic reference simulation (e.g. a simulation without any INP).

Change to paper: The following lines have been amended to make the use of s and r for the purposes of this paper clearer:

"The cloudy regions contribution, i.e. the difference in outgoing radiation between two cloudy regions due to changes in cloud albedo or thickness ignoring any changes in cloud fraction, $(\Delta Rad_{REFL})$ to a domain radiative difference **between a sensitivity simulation (s) and a reference simulation (r) (s – r)** is calculated using Eq. (1).

$$\Delta Rad_{REFL} = cf_r \times \Delta Rad_{cl} \qquad\qquad (1)$$

where $cf_r$ is the cloud fraction of simulation r and $\Delta Rad_{cl}$ is the change in outgoing radiation from cloudy areas only between simulations (s – r). **The reference run (r) in Sections 3.1 – 3.4 refers to the NoINP simulation while the sensitivity run (s) are simulations which include an INP parameterisation. In Section 3.5, the reference run (r) refers to a simulation which has no representation of SIP and the sensitivity run (s) to a simulation which includes SIP due to the Hallett Mossop process.**"

Line 201: I am not sure if I followed equation 2 correctly, but could Δcf in this equation be replaced with (Rads,cl – Rads,cs) since you are looking at the contributions of cloud fraction to radiation between simulation s and r? If this is correctly understood, I would replace the Δcf with (Rads,cl – Rads,cs), to make it easier to follow equation 2. Or define Δcf more explicitly than has been done in line 203.

Reply: This equation calculates the change in outgoing radiation due to the increase or decrease in domain cloud fraction and as such calculates the change in outgoing radiation that occurs when an area in simulation s that was clear sky becomes cloudy in simulation r (or vice versa). It therefore calculates the average difference between cloud and clear sky in simulation s and multiplies this by the proportion of the domain that the change occurs in (the cloud fraction change between simulation s and r, or $\Delta cf$). This therefore assumes that the outgoing radiation from cloudy and clear regions in simulations s and r is the same. We know this assumption is false from the result of Equation 1 which is why we calculate the interaction effect (i.e. what the contribution of changes in cloud reflectivity or thickness in regions that have become cloudy is) to domain outgoing radiation from cloudy and clear sky regions in Equation 3. However the contribution of $\Delta Rad_{INT}$ calculated in Equation 3 was negligible and therefore we are happy that Equation 2 captures the majority of the change in outgoing radiation due to cloud fraction change.

Change to paper: We have altered the description of Equation 2 as follows to make clearer what the equation defines and that $\Delta cf$ refers to the change in domain cloud fraction between simulations s and r:

*"**The contribution of cloud fraction changes, i.e. the change in radiation that can be attributed to an area of clear sky in simulation s becoming cloudy in simulation r or vice versa, to the total change in domain outgoing radiation** ($\Delta Rad_{CF}$) is calculated using Eq. (2).*

$$\Delta Rad_{CF} = \left(Rad_{r,cl} - Rad_{r,cs}\right) \times \Delta cf \qquad (2)$$

*Where $Rad_{r,cl}$ is the mean outgoing radiation from cloudy regions in simulation r and $Rad_{r,cs}$ is the mean outgoing radiation from clear sky regions in simulation r and $\Delta cf$ is the difference in **domain** cloud fraction between simulations s and r (s-r)."*

Line 239: If the evolution of the clouds are not being discussed further, then no need to show the plots in figure 2e, f,g and h. The most useful plots are a-d since they compare the satellite with the model. Since there are no comparisons to the satellite for the other times, it doesn't give much of a validation. Also to show that the model produces a complex realistic cloud field can be demonstrated with Fig2c and 2d, so that c-h are not necessary.

Change to paper: As suggested.

Line 242: Same comment as above with Fig A2, panel c, d, e and f are not necessary.

Change to paper: As suggested.

Line 255: "flow" should be "flown"

Change to paper: As suggested.

Line 272: In the noINP case, can ice crystals that formed via homogeneous freezing, fall to lower levels and initiate secondary ice processes?

Reply: Yes, this is possible.

Change to paper: We have added the following sentence to clarify this:

*"Ice crystals formed via homogeneous freezing can sediment to lower levels and initiate ice production via the Hallett Mossop process."*

Lines 281 to 283: similar to comment above, the comparison of noINP to simulations with INP parameterization demonstrates an enhancement in outgoing radiation for D10 and A13. Can the authors clarify here that noINP simulation excludes any contribution of SIPs that could result from ice settling from higher altitudes to warmer regions of the clouds? Or is such a contribution included in this assessment?

To me this seems possible to diagnose in the model, if the assessment of precipitation evaporating results in higher humidity such that the LWP due to increase cloud droplet number. Then why isn't the possibility of ice crystals or snow settling through the clouds at Hallet-Mossop relevant temperatures allowed to produce secondary ice?

Reply: The Hallett Mossop process can occur and produce ice due to ice settling in the NoINP simulation, as can be seen in Figure A4a.

Change to paper: We have added the bold sections in the following sentences to clarify this:

*"We first examine the effect of INP parameterisation on the outgoing radiation relative to the simulation where the only source of primary ice production was through homogeneous freezing (NoINP). **Cloud ice formed via homogeneous freezing can sediment to lower levels and initiate ice production via the Hallett Mossop process.** When contrasting the effect of different INP parameterisations in Sect 3.1-3.4, the Hallett Mossop process was always active **including in the NoINP simulation**. As stated in Sect. 2.1.3, the radiation code is represented by the Suite Of Community RAdiative Transfer codes based on Edwards and Slingo (SOCRATES) (Edwards and Slingo, 1996; Manners et al., 2017), and responds to changes in cloud droplet number and cloud droplet, ice crystal and snow mass. The results detailed below relate to either the domain-wide properties or all in-cloud regions within the domain. This means that the results describe the direct and indirect changes**, for example changes to the Hallett Mossop ice production,** occurring due to **the presence of** INP across all cloud present in the domain, including low-level liquid clouds, mixed-phase clouds without a convective anvil and very deep convective clouds with an anvil."*

Line 285-286: "Radiative changes from the NoINP simulation to the inclusion of INP are caused mainly by .."

It is not cleat to me what is meant by this statement. I think it means the difference in radiation between the NoINP and the INP simulations mostly arises from the changes in outgoing SW radiation, but if that is the case why not state it more simply, i.e. the difference in radiation between NoINP and INP simulations …

Change to paper: As suggested.

Lines 291 – 292: The reported comparatively small change in TOA radiation when SIP is active relative to when it is inactive mentioned here, could this be because in the homogeneous freezing run SIP is by default assumed to be absent (i.e. settling of crystals to warmer Hallet-Mossop regions for secondary ice is not represented in the model)?

Reply: The settling of crystals to warmer Hallet-Mossop regions for secondary ice is represented in the model and is allowed to produce secondary ice as can be seen in Fig A4a.

Change to paper: We have added the bold in the following sentence to make this clear:

*"Bear in mind that SIP was active (SIP_active) in the simulations summarised in Fig. 4a**, including in the NoINP simulation in which the Hallett Mossop process can be initiated by settling ice crystals,** indicating that these cloud systems are sensitive to INP even in the presence of SIP."*

Line 293: The relationship between the slope of the parameterization and the outgoing radiation is implicit and not explicit. So stating that it is the key determinant seems very strong here, without clarifying that the slope is

representative of the INP concentration as a function of T (Fig 1). It is the T at which a certain proportion of INPs are active is they key determinant and the physical reason for the strong influence on available supercooled liquid to be transported to higher altitudes of the MPC region. Could this relationship be clarified to invoke the temperature dependence rather than just stating the slope of the parameterization is the key determinant?

Reply: This comment is correct in that it is the INP concentration as a function of T that is important. We have termed this dependence of INP number concentration on temperature as the slope for simplicity as stated in the introduction "In particular, we examine the importance of the dependence of INP number concentration on temperature, referred to as INP parameterisation slope herein, as a major factor that determines cloud properties."

Change to paper: We have altered this line as follows to make clearer that it is the temperature dependent concentration that is important:

*"The slope of the INP parameterisation **(i.e. the dependence of INP number concentration on temperature)** is a key determinant of the outgoing radiation."*

Line 303-304: This sounds counterintuitive to me because compared to the NoINP simulation, the increase in INP should increase the ICNC and decrease the cloud lifetime or outgoing SW (reflectivity) compared to otherwise what would be a more reflective liquid cloud with less ice. However further down the authors do explain why they observe this, because the liquid water path increases in the warmer part of the cloud which increases the outgoing SW. However, I find this assessment to be biased, without accounting for potential SIP in the NoINP simulation due to settling ice crystals.

Reply: We have addressed the issue of SIP in the noINP case above. To be clear, SIP due to settling ice crystals is possible in the NoINP simulation, as can be seen in Figure A4a. In the Hallett Mossop region, the ICNC is lowest in the NoINP simulation (see Figure AC1 below) because of the reduced production of ice by SIP. However, in all other regions of the cloud, including the upper mixed phase temperatures, ICNC is highest in the NoINP simulation due to the enhanced ice production by homogeneous freezing (Figure A4b), and subsequent settling of the ice crystals.

[Figure]

Figure AC1. In-cloud profile of ice crystal number concentration in the parameterisations that included SIP by the Hallett Mossop process.

Line 309/Figure 4b: I would have expected the water path due to snow to decrease because snow should leave the cloud faster than ice crystals? So why is the water path due to ice crystals decreasing? Or does it have to do

with the categorisation of when a hydrometeor is considered a snow flake vs. an ice crystal in the model? Also, I would have expected that the increase in water path should be the lowest for the M92 and not for the D10, since the M92 has the shallowest slope?

Reply: The water path of ice crystals (cloud ice) decreases (in all except A13) when INP are included because in the NoINP simulation, more homogeneous freezing occurs and these ice crystals can settle through the cloud. In the simulations with INP, snow production is enhanced in the mixed-phase region (Figure A4c) due to enhanced accretion, riming and aggregation which increases the in-cloud snow water path. This increase is large enough to compensate for any decreases in snow mass that may occur due to increased settling. The changes in snow water path do not correspond to INP parameterisation slope because the increase in snow mass is relatively similar between all parameterisations (Figure A4c).

Lines 313-320: If the precipitation is increased, then the snow and graupel should not be part of the cloud anymore and thus not contribute to the increased reflected shortwave radiation. If the increased condensate is falling as precipitation such that it is resulting in an increased humidity thus increasing the LWP from an increase in cloud droplets, how can it also contribute to increasing the outgoing shortwave in the cloud? It should either be classified as increased precipitation below cloud or increased condensate in-cloud leading to outgoing shortwave.

Reply: The in-cloud snow and graupel mass is enhanced significantly from the NoINP simulation (Fig A4c, d), allowing a proportion of snow and graupel to both remain 'in-cloud' and precipitate out of cloud to cause increased humidity at low cloud levels. Once the snow and graupel precipitate out of cloud below the melting temperature, they will be classified as rain in the model. Rain is excluded from the cloud mass threshold and thus hydrometeors that have precipitated out of cloud and melted are not included in the 'in-cloud' values.

Change to paper: We have added the bold text to the following sentence to clarify this:

*"At the same time, the enhanced production of relatively heavy snow and graupel increases precipitation which **on melting to form rain below the freezing level and subsequent** evaporation below 4 km, reduces out-of-cloud temperature and increases relative humidity (Fig. A5a, b)."*

Lines 321 – 334: This explanation makes more sense and sounds stronger and more convincing to me than the explanation in lines 308 to 320. Perhaps it would be better to shift the order of these paragraphs and explain the higher outgoing shortwave by the increased CDNC at lower altitudes due to increased LWP from lower freezing rates arising from steeper INP parameterizations (which imply very little het. freezing at small supercooling).

Reply: The lines in question relate to the difference in outgoing radiation from cloudy regions between simulations using different parameterisations and the impact of the temperature dependence of INP number concentrations. However, we feel it is logical and important to first address the reason for the change in outgoing radiation from the NoINP simulation before addressing the difference between different INP simulations.

Line 331: clarify statement more, I suggest (italics is suggested part) something like "…due to lower rates of heterogeneous freezing at the mid-bottom region of the mixed-phase cloud (lower supercooling, Fig. 1) and SIP at …"

Change to paper: As suggested.

Line 339: clarify by inserting "cloud fraction due to" i.e. sentence should read

"…offset somewhat by decreases in the cloud fraction due to homogeneous freezing in the ~10 – 14 km regime (Fig 6a)"

Change to paper: The sentence has been altered as suggested with one minor difference: "offset somewhat by decreases in the cloud fraction due to **reduced** homogeneous freezing in the ~ 10 - 14 km regime (Fig. 7a)"

Line 358-360: If this is true, (and it sounds like a good explanation), shouldn't the decrease in outgoing LW shown in Figure 4a be the highest for A13 and not for C86. Because A13 results in the highest number of ICNC at the top of the MPC region and in the homogeneous freezing region therefore should trap most of the outgoing LW radiation thus giving the most decrease in the outgoing LW.

Reply: The sentence in question refers to the ratio of ice crystals to snow and graupel which affects the cloud fraction, not their absolute values which effects the cloud thickness and thus the outgoing longwave. The decrease in outgoing longwave radiaiton would be greatest for A13 if the only thing contributing to outgoing longwave was ice crystal number or ice crystal mass, however the change in outgoing longwave from cloudy regions is due to changes in all of snow, ice crystals and cloud droplets. The decreases in longwave are predominately due to increases in total waterpath, and thus cloud thickness, from cloud droplets, ice crystals and snow particles. C86 has the greatest combined water path from cloud droplets, ice crystals and snow particles (Figure 4c) and thus the largest decrease in outgoing longwave from the NoINP simulation.

Line 379:. Change to "It has been argued that the observed (or derived) primary ice particle production rate…". Otherwise, the statement is false, because if the primary production rate is high, the secondary ice production (H-M process) would be low but still present, primary ice production would in fact dominate cloud properties.

Change to paper: As suggested.

Line 384/ line 219/lines 440-445: Most relevant comment. You show that higher primary ice production rates in the temperature range between 253 and 238 K, e.g. in A13, have a large impact on the total on top of atmosphere outgoing radiation, yet you exclude SIP which are active at colder temperatures than 265 K. Can you elaborate on the expected impact/uncertainty in your results and concluding statement, (SIP is less important than primary ice production) stemming from your simplification that SIP is only including Hallett Mossop process? Please justify why your concluding statement is valid.

Reply: It is correct that we have only modelled Hallett Mossop SIP and have made this much clearer in the revised paper (i.e. rather than referring generally to SIP, we refer to Hallett-Mossop SIP). However, it is also true that other theoretical SIP mechanisms are unquantified and it has not been clearly shown that they need to be included in models, beyond specific cases. As understanding of other potential SIP mechanisms emerges they should be included and assessed in studies like this one in future research.

Change to paper: We have clarified wherever relevant that the only SIP included in our simulations is the Hallett Mossop process and caveated if necessary that the results may be different if other mechanisms of SIP were included.

Line 383-384: This is an important outcome of the study, but should be caveated with the notion that other possible known and unknown SIPs are not considered. The authors in part do that by acknowledging in parenthesis that the SIP considered is the H-M, but I think they could go one step further in saying that this could change with more parameterizations and quantification becoming available for lower temperatures where SIP becomes important say below 265 K (Lauber et al., 2018).

Change to paper: As suggested.

Page 14 (line 385). An average impact comparison in the text might be supportive for the reader (e.g. mean over all parameterizations total INP impact 9.8 W/m2 to total SIP impact 2.7 W/m2)

> Change to paper: As suggested.

Line 453-455: A possible explanation for this statement should be given here in in the conclusions, since this is an important point or outcome of the study. Have the other studies that are mentioned in these lines also only tested the influence of the Hallet-Mossop process? If not, this should be clarified. Further, since this has evaluated the influence of SIP due to the HM process, it should be stated here in the conclusions.

So this conclusion is true, when the SIP being considered is HM. But it remains open if the conclusions would still hold if freeze shattering and other mechanisms (e.g., as described in Lauber et al., 2018) were included in the models.

> Reply: The papers cited either explicitly tested the Hallett Mossop process or inferred using observational data that INP concentrations were relevant up to a threshold needed to initiate the Hallett Mossop process.

> Change to paper: The section has been modified to make clear that only the Hallett Mossop process was included.

Line 496: In addition to Holden et al. (2019) the authors could add Coluzza et al. (2017) and Kanji et al. (2017) since that lack of knowledge on what constitutes the identity of an active site was already discussed in these publications.

> Change to paper: As suggested.

Line 497: The last statement here has surely been mentioned before by other authors in numerous publications. While it is valuable that the authors also come to this conclusion (need for INP measurement across the entirety of the MPC regime), this study is not the first to recommend such an outlook and the sentence can be modified to say "..as reported before by XX.."

> Change to paper: As suggested the line in question has been altered to "This work demonstrates the importance of solving these problems and measuring INP number concentrations across the entirety of the mixed-phase temperature spectrum**, as has been demonstrated in previous work (e.g. Liu et al., 2018; Takeishi and Storelvmo, 2018)**."

Figure 2: Should there not be a "radiation" in the color bar caption, e.g. Outgoing OA longwave radiation (W m-2)?

> Change to paper: As suggested. The same change was made in Figure A1.

Figure 5: Colour legend is missing. I suggest adding it here even if it is the same as previous figures for ease of reading.

> Change to paper: As suggested.

Appendix A, page 34 (Figure A1). The three kind of blue lines are not easy to distinguish from the black line.

> Change to paper: The blue lines in the plot in question (Now Figure 1b,c) have been changed to red.

Appendix A, page 34 (Figure A1). What does the unit of / 108 m-3 mean? Is it two particles per 108 m-3 of volume?

> Change to paper: Changed to '$10^8\,m^{-3}$ ' to reflect $10^8$ particles $m^{-3}$ of volume (now Figure 1b). There was a similar error in Figure 7b y-axis units changed from /$10^{-5}\,m^{-3}$ to '$10^5\,m^{-3}$'.

Figure A3 panel a: what are the regions filled with black colour? Could the colors be changed so that the contrast between green and blue is better visible? If black is just the border of the bars, I suggest removing the borders since it reduces clarity of the plot suggesting that there is a third colour.

Change to paper: As suggested, the black outline has been removed.

Line 810: Histograms should be singular not plural.

Change to paper: As suggested.

References

Coluzza, I., Creamean, J., Rossi, J. M., Wex, H., Alpert, A. P., Bianco, V., Boose, Y., Dellago, C., Felgitsch, L., Fröhlich-Nowoisky, J., Herrmann, H., Jungblut, S., Kanji, A. Z., Menzl, G., Moffett, B., Moritz, C., Mutzel, A., Pöschl, U., Schauperl, M., Scheel, J., Stopelli, E., Stratmann, F., Grothe, H., and Schmale, G. D.: Perspectives on the Future of Ice Nucleation Research: Research Needs and Unanswered Questions Identified from Two International Workshops, 8, doi:10.3390/atmos8080138, 2017.

Holden, M. A., Whale, T. F., Tarn, M. D., O'Sullivan, D., Walshaw, R. D., Murray, B. J., Meldrum, F. C., and Christenson, H. K.: High-speed imaging of ice nucleation in water proves the existence of active sites, 5, eaav4316, doi:10.1126/sciadv.aav4316, 2019.

Kanji, Z. A., Ladino, L. A., Wex, H., Boose, Y., Burkert-Kohn, M., Cziczo, D. J., and Krämer, M.: Overview of Ice Nucleating Particles, in: Ice Formation and Evolution in Clouds and Precipitation: Measurement and Modeling Challenges, 2017.

Lauber, A., Kiselev, A., Pander, T., Handmann, P., and Leisner, T.: Secondary Ice Formation during Freezing of Levitated Droplets, 75, 2815-2826, doi:10.1175/JAS-D-18-0052.1, 2018.

Peckhaus, A., Kiselev, A., Hiron, T., Ebert, M., and Leisner, T.: A comparative study of K-rich and Na/Ca-rich feldspar ice-nucleating particles in a nanoliter droplet freezing assay, Atmos. Chem. Phys., 16, 11477-11496, doi:10.5194/acp-16-11477-2016, 2016.

**Replies to review RC2**

Review "The nature of ice-nucleating particles affects the radiative properties of tropical convective cloud systems" by R. E. Hawker, et al.

Thank you for your constructive comments on our paper.

This modeling study investigates the effects of ice nucleating particles (INP) on the radiative properties of tropical convective cloud system. Several widely used INP parameterizations are used in the model, and interestingly the slope of the INP temperature dependence is found to play a key role in the INP effects. The effects of secondary ice production (SIP) are studied, which demonstrate the important role of INP nature for the SIP effects. The study examines the difference aspects of cloud microphysical properties and cloud fraction to understand the INP effects on radiative forcing. Generally, the research topics of INP and SIP and effects on convection are interesting and the results are novel. The manuscript may be accepted for publication after addressing my comments.

Main comments

1. Adding a schematic to illustrate how different slopes of INP temperature dependence could affect cloud microphysics at different vertical layers of convective clouds. That will help readers to better understand the interactions.

Change to paper: As suggested, a schematic has been added (Fig. 9, shown below)

[Figure]

**Figure 9. Schematic of the main effects of INP parameterisation slope (i.e. a steep (a) or shallow (b) temperature dependence of INP number concentrations).**

2. A better description of model configuration such as model domain, and initial thermodynamical profiles, is required.

Change to paper: As suggested. Fig. 1a (shown below) has been added to the paper to better illustrate the domain location and how the global model is used to initiate the nested domain. Fig. A1a and b (now Fig. 1b, c, shown below) have been moved to the main paper so that the initial aerosol profiles are clearer to the reader. More detail has been added to the reference to the global model (from "The Met Office global model is used to initialise the nested simulation" to "A global model simulation (UM vn 8.5, GA6 configuration, N512 resolution (Walters et al., 2017)) is used to initialise the nested simulation").

[Figure]

**Figure 1. Modelled domain location and resolution details (a), observed (black line) and modelled (red lines) aerosol concentrations (b), and mean modelled domain mean temperature and relative humidity profiles (c). The observed aerosol profile shown in b was measured using the Passive Cavity Aerosol Spectrometer Probe (PCASP) which captures aerosols between 0.1 and 3µm in size. The insoluble aerosol profile shown in b is extracted from a regional UM vn 10.3 simulation (8 km grid spacing, CLASSIC dust scheme). The modelled aerosol profiles are applied throughout the regional domain shown in a at the start of the simulation (00:00 21st August 2015) and at the boundaries throughout. INP concentrations in the D10, N12 and A13 simulations are linked to the insoluble aerosol profile shown in b. The image shown in (a) are moderate resolution imaging spectroradiometer (MODIS) Corrected Reflectance imagery produced using the MODIS Level 1B data and downloaded from the NASA Worldview website.**

Change to paper: As a result of the above two changes the numbers of the other figures in the paper have been changed.

Minor comments

1. Line 45: "..INP number concentrations can vary by as much as six orders of magnitude at any one temperature". Can you add some reasons to explain this large variability?

Change to paper: The sentence in question has been altered to

*"Measurements indicate that INP number concentrations can vary by as much as six orders of magnitude at any one temperature **due to variations in, for example, aerosol source, chemical or biological composition and surface morphology** (DeMott et al., 2010; Kanji et al., 2017).*

2. Line 56: "...global models based on known INP-active materials show reasonable skill in simulating global INP concentrations (Vergara-Temprado et al., 2017)." There are also other studies that can be cited here: e.g., Shi and Liu (2019), GRL, compared the modeled INPs with observations.

Change to paper: As suggested

3. Line 81: "and droplets can freeze homogeneously below around -33°C". Does your model represent the homogeneous freezing of aerosol droplets below around -37°C?

No, it represents the freezing of cloud droplets only.

4. Lines 93-103. You can add one sentence here to introduce your work on SIP effects and dependence on INP parameterizations.

Change to paper: As suggested.

5. Section 2.1.1. Please add a figure in main text showing the model domains and how different models, cloud microphysics, and aerosols are applied to these domains.

Change to paper: As suggested, a new figure has been added showing the model domains. Former Fig. A1a and b was combined with this new figure becoming Fig. 1b and c.

6. Line 131. "five hydrometeor classes (cloud droplets, rain droplets, ice crystals, graupel, snow)". It would be clearer to use "cloud ice" to replace "ice crystals" since ice crystals can contain snow and graupel particles. Also changes the words in text.

Change to paper: We have clarified in this sentence that throughout the paper ice crystals refers to cloud ice.

7. Line 149. You can make it clear the insoluble aerosol is dust which is used in N12 and A13 parameterizations.

Change to paper: We have altered "insoluble aerosol" to "insoluble **dust** aerosol".

8. Line 158: "predict an ice production rate via heterogeneous freezing". These ice nucleation parameterizations (M92, N12, D10) only predict INP number concentrations. How do you calculate the ice production rate?

Reply: CASIM examines the gridbox temperature, cloud number, ice number and, in the case of D10, N12 and A13, the insoluble aerosol concentrations. If the cloud number is above a minimum threshold and the temperature is below 0°C, it calculates the available INP. It then subtracts the existing ICNC in the gridbox from this calculated INP number concentration to calculate the ice crystals produced by heterogeneous freezing giving an ice production rate. The number of ice crystals produced by heterogeneous freezing is output as a 3D diagnostic from the model.

Change to paper: We have altered the sentence as follows to make it clearer what else other than insoluble aerosol number concentration is used to calculate the rate of ice production.

*"The INP parameterisations inspect the conditions **(temperature, cloud droplet number, ICNC)** and aerosol concentrations within a gridbox and use that information to predict an ice production rate via heterogeneous freezing."*

9. Line 159. Do you also represent the CCN wet scavenging?

Reply: No the aerosols are not depleted by cloud droplet activation or ice nucleation in these simulations. We have clarified this in the second paragraph of Section 2.1.2.

10. Line 223. It would be good to note that these INP parameterizations only applied to certain temperature ranges and cannot reliably extrapolate to temperatures outside the range.

Change to paper: As suggested, we have added the following to make this clear:

"*It should be noted that all parameterisations tested in this work were developed between specific temperature ranges and extrapolation beyond these temperatures adds uncertainty. However, for the purposes of this paper and to allow a direct comparison between parameterisations, all parameterisations have been applied between 0 and -37°C.*"

11. Section 3.5. Line 384. How do other SIP mechanisms impact your results here in this section?

Reply: Other SIP mechanisms are not included in the model simulations. Their inclusion may alter the results presented which only relate to the Hallett Mossop process because it is the most well-known and best quantified SIP mechanism.

Change to paper: The paragraph in question has been altered as follows to highlight the limitations and caveats of not including other SIP mechanisms:

"*We find that the microphysical and radiative properties of the cloud field depend strongly on the properties of the INP even **when SIP due to the Hallett-Mossop process occurs**. Furthermore, the effect of including SIP on daylight domain-mean outgoing radiation varies between –2.0 W m$^{-2}$ and +6.6 W m$^{-2}$ (Fig. 4b), showing that **the presence of the Hallett Mossop process** has a smaller effect than the INP parameterisation **and that the sign and magnitude of this effect** depends on the INP parameterisation. Therefore, rather than primary ice being simply overwhelmed by SIP, it actually determines how SIP affects cloud microphysics. **Other mechanisms of SIP have been proposed (Field et al., 2017; Lauber et al., 2018) and the impact of INP on cloud properties in the presence of these mechanisms, particularly those present at temperatures below 10°C such as droplet shattering (Lauber et al., 2018), should be tested in future but this was beyond the scope of the present study.**"*

12. Line 399-407. As commented above, a schematic showing the interactions and mechanisms would be helpful for the readers.

Change to paper: As suggested, a schematic has been added to the paper (Fig. 9)

13. Line 412. Since CASIM is a two-moment cloud microphysics scheme, why are ice and snow particle numbers not used in the radiation scheme?

Reply: The SOCRATES radiation scheme is the radiation component of the Met Office Unified Model, and has not yet been fully coupled to the CASIM microphysics module. As such ice and snow particle numbers, and thus size, have not yet been included in the radiation calculations. The incorporation of ice and snow particle numbers into the SOCRATES scheme is a desirable feature for future model development but was beyond the scope of this paper.

14. Line 415. It is not correct to state: "climate models do not typically represent ICNCs". Please remove.

Change to paper: Altered as follows to be more specific that we are referring only to the representation of ICNC in radiation calculations in climate models:

"*However, our results are still very relevant for climate model simulations as **climate models** typically employ single moment microphysics schemes that do not explicitly predict ICNC for the purposes of inclusion in radiation calculations** and have frequently been shown to poorly represent ice crystal mass concentrations (Baran et al., 2014b; Waliser et al., 2009).*

---

## Author Response (AR2)

Dear Martina,

Thank you for your continued work on our manuscript. Please find below details of the manuscript changes in response to the second round of reviewer comments.

Best wishes,

Rachel Hawker.

**Replies to Report #2**
* * *
In this work the authors investigate the effect of the parameterization of ice nucleation particles (INP) on the development of convective clouds and their radiative properties. The authors perform simulations of a tropical convective event using a set of different INP parameterizations with diverse dependency on aerosol concentration and temperature. Their work demonstrates that ice initiation may affect strongly the radiative properties of these systems, and in contrast to previous work, such an effect is not buffered by secondary ice production. This is a well-written paper, relevant to the atmospheric community. I'd recommend its publication in ACP after a few minor clarifications.

Thank you for your work and your helpful comments on our paper.

**Comments.**

This paper has gone already through a review process, which resulted in an improved work. Hence, I just have a minor comment. The authors must emphasize that the INP parameterizations refer only to immersion freezing and provide some information (probably in section 2.1.2) on the other ice formation mechanism in the model. I am referring specifically to contact ice nucleation in the convective cloud, and the in situ formation of cirrus by heterogeneous ice nucleation by deposition, and, by the immersion and homogeneous freezing of haze droplets at T < 235 K. The latter could be quite relevant since in situ cirrus may interact with anvil particles and may either exacerbate or negate the observed radiative impact.

Change to paper: The following text has been added to Section 2.1.2: "The INP parameterisations tested in this study represent only immersion freezing. Heterogeneous ice nucleation by deposition and contact nucleation are not represented. Other mechanisms of heterogeneous ice formation should be tested and included in future studies but was beyond the scope of this work. However, immersion freezing is expected to be the dominant mechanism of heterogeneous ice formation in convective clouds (Ansmann et al., 2008; De Boer et al., 2011; Kanji et al., 2017) and therefore the simulations presented here should capture the majority of heterogeneous ice nucleation relevant for cloud properties. Immersion and homogeneous freezing of haze droplets are not represented, but it is unlikely that they contribute significantly to ice crystal number concentration in the main anvil cloud derived from mixed-phase cloud regions. However, the importance of these mechanisms on anvil cloud properties should be investigated in future work."

**Replies to Report #3**
* * *
In this study, the authors test the sensitivity of a mix of Eastern Atlantic Tropical cloud types to INP parametrizations with different temperature dependencies as well as the Hallett-Mossop process. They show that the slope (temperature dependence) of INP parametrizations has a significant impact on the TOA outgoing radiation for the clouds studied. Additionally, even in the presence of the Hallett-Mossop process, the INP parametrizations still produce significant results, indicating that correctly parameterizing the temperature-dependence of INPs is important to accurately simulate the radiative properties of clouds.

The manuscript is well written and the explanations are well thought through and explained nicely. The authors have also done a nice job responding to the previous reviewer comments. Nevertheless, I still think some points could be better explained as well as have a few additional comments listed below. The line and figure numbers correspond to the line numbers in the track changes version of the manuscript with the reviewer comments and author responses.

**Thank you very much for your constructive comments on our paper.**

**General/minor comments:**

As previously mentioned, it is not immediately clear which hydrometeor classes are able to gain mass through riming and therefore produce secondary ice through the Hallett-Mossop process (HMP). Please clarify this as in several places, ice crystals are mentioned specifically but no other ice phase hydrometeor classes (snow and graupel) are mentioned. In response to a previous reviewer comment it is stated that ice crystals are meant to represent all cloud ice but for clarity it might be easier to replace ice crystals with ice phase hydrometeors or something more inclusive in the text when referring to the HMP. See specific line numbers below.

Reply: Ice crystals, snow and graupel can all gain mass through riming. However, only riming from snow and graupel species produce splinters via the Hallett-Mossop process. This is the case because the majority of riming occurs on snow and graupel particles with a negligible contribution from cloud ice or small ice crystals which are converted to graupel before substantial riming, and therefore rime-splintering, can occur. We appreciate this was not clear previously and have clarified this in the locations in the text highlighted in your minor comments below. When we refer to ice crystals initiating the Hallett-Mossop process, we are referring to an indirect initiation by allowing the earlier formation of snow and graupel in the cloud, and therefore driving the subsequent riming and splinter production that follows snow and graupel formation.

As previously mentioned by a reviewer, it is stressed that INPs are not scavenged in the text. But what does this actually mean? In the response, the authors state that as long as a threshold number of droplets is present and based on the ICNC already present and T, a freezing rate is calculated. This makes me wonder if the ICNC in a grid box is already above the predicted number of ice crystals from a parametrization due to settling (lifting) of ice from above (below), is no new ice formed? Please clarify this and also discuss what impacts this might have on the influence of the INP parametrizations used and the resulting TOA radiation if this is the case.

Reply: The reviewer is correct that if the ICNC in a grid box is already above the predicted number of ice crystals from a parameterisation due to settling (lifting) of ice from above (below) no new ice is formed. This is the main caveat of this study resulting from the exclusion of aerosol scavenging. The INP parameterisation computes the change in ICNC resulting from INP at the timestep after the ice nucleation occurs. This means that in the case of strong updraft, such as in the deep convective core, it is our expectation that the representation of heterogeneous freezing accounts for scavenging in a primitive way and should come relatively close to what we would expect if the background aerosol was being removed by ice nucleation. In the case of strong

sedimentation of ice crystals from above, the lack of scavenging would lead to an underestimation of heterogeneous freezing. However, as the terminal velocity of ice crystals is very low, this would not be expected to play a large role except in very weak updrafts outside of the deep convective core. Scavenging can have counterintuitive effects on the cloud properties making speculation on its effect difficult but we agree that this is an important question to address in future work.

Change to paper: The following bold text has been added to the limitations section of the paper (Section 4) to make clearer the limitations of excluding aerosol scavenging: "The effect of INP parameterisation choice on convective cloud field properties should also be examined with the inclusion of aerosol scavenging but this was beyond the scope of this study. *Aerosol scavenging would allow the aerosol number concentration to be reduced by cloud droplet activation and the number of dust particles within cloud droplets to be tracked and depleted when frozen heterogeneously. In the simulations presented here, the heterogeneous freezing rate is calculated using the interstitial aerosol number concentration and the ICNC of the gridbox in question meaning that ice crystals advected into the gridbox will reduce the heterogeneous nucleation rate even if they were frozen elsewhere in the domain."*

Similarly, a threshold number of cloud droplets is used. But what is this number? Is it large enough so that there are always enough cloud droplets such that the number of formed ice crystals does not exceed the number of cloud droplets? Please clarify this and discuss the impacts on the results if this is not the case.

Reply: The gridbox conditions (temperature, cloud droplet number, ICNC, and aerosol concentrations) are inspected and the number of heterogeneously formed ice crystals is calculated if the cloud droplet number concentration exceeds 1.0e-6 kg-1. If the number of heterogeneously formed ice crystals calculated by the parameterisation exceeds the cloud droplet number concentration, all the cloud droplets in a gridbox are frozen. The number of formed ice crystals cannot exceed the number of cloud droplets.

"Hallett Mossop" should be changed to "Hallett-Mossop" throughout the text.

Change to paper: As suggested.

It is not always clear if the results are referring to the TOA changes in radiation over the entire daylight period (10-17 UTC) or the entire simulation (10-24). Please make that clearer in the text and the figure captions rather than just mentioning it in the Methods.

Change to paper: The following figure captions have been altered to make clear what time period the results depict:

Caption Figure 4: "*Effect of INP and secondary production on top of atmosphere (TOA)* **daytime (10:00-17:00 UTC)** *outgoing radiation.*"

Caption Figure 5: "INP and TOA outgoing daytime (10:00-17:00 UTC) radiation from cloudy regions."

Caption Figure 6: "Outgoing **daytime (10:00-17:00 UTC)** radiation from cloudy regions and INP parameterisation slope."

Caption Figure 8: "Vertical composition of cloud. 2D distribution of cloud type at 20:00 for all six SIP\_active simulations (a-f), as well as anvil and domain cloud fraction change **(10:00-24:00 UTC)** due to INP"

Caption Figure A4. "Effect of INP on domain-mean TOA outgoing daytime (10:00-17:00 UTC) shortwave and longwave radiation."

Additionally, the following sentences at the start of the relevant results sections have been altered to make clear that they are discussing the daytime outgoing radiation:

Start of Section 3.1: "We first examine the effect of INP parameterisation on the **TOA** outgoing **daytime** (10:00-17:00 UTC) radiation relative to the simulation where the only source of primary ice production was through homogeneous freezing (NoINP)."

Start of Section 3.2: "Here we discuss the changes in **daytime** outgoing radiation from cloudy regions only due to INP parameterisation choice."

Start of Section 3.3: "Overall cloud fraction is increased by INP for all INP parameterisations and these increases in cloud fraction contribute about as much to changes in domain-mean **daytime** radiation as the changes in outgoing radiation from cloudy regions"

I find the equations nice to explain the different calculations that have been conducted. However, there is no return to the acronyms used in the equations and therefore it is not immediately clear which values are calculated with which equation. Consider integrating the equation acronyms into the text and figures. Although, in general I find the text explanations quite clear, but also mentioning the acronyms, especially in the figures might make things clearer as how things were calculated.

Change to paper: The following figure captions have been altered to refer back to the terms calculated in Section 2.1.3.:

Caption Figure 4: "In (a), the change from the NoINP simulation is shown (INP - NoINP) with SIP active. In (b), the change from SIP\_active to SIP\_inactive is shown (SIP\_active – SIP\_inactive). A positive value indicates more outgoing radiation when INP or SIP are active. In (a) and (b), the relative contributions of changes in outgoing radiation from cloudy regions (left) (i.e.  $\Delta Rad_{REFL}$  from Eq. (1)) and cloud fraction (middle) (i.e.  $\Delta Rad_{CF}$  from Eq. (2)) to the total radiative forcing (right) (i.e.  $\Delta Rad_{s-r}$  from Eq. (5) with simulation s referring to simulations with INP active in (a) and to the SIP\_active simulations in (b) and simulation r referring to the NoINP simulation in (a) and to the SIP\_inactive simulations in (b)) are shown (calculation described in Sect. 2.1.3)."

Caption Figure 5: "Absolute change in outgoing shortwave, longwave and total radiation from cloudy regions relative to the NoINP simulation (i.e.  $\Delta Rad_{cl}$  used in Eq. (1)) (a)"

Caption Figure A4: "Effect of INP on domain-mean TOA outgoing daytime (10:00-17:00 UTC) shortwave and longwave radiation. The change from the NoINP simulation is shown (INP - NoINP). A positive value indicates more outgoing radiation when INP are present. The contributions of changes in outgoing radiation from cloudy regions (left) (i.e.  $\Delta Rad_{REFL}$  from Eq. (1)) and cloud fraction (middle) (i.e.  $\Delta Rad_{CF}$  from Eq. (2)) to the total radiative forcing (right) (i.e.  $\Delta Rad_{s-r}$  from Eq. (5) with simulation s referring to the simulations with INP active and simulation r referring to the NoINP simulation) are also shown (calculation described in Sect. 2.1.3)."

It is stated in the manuscript that there was little sensitivity to the time frame (day versus day and night) considered for the impacts on TOA outgoing radiation. However, I find this quite surprising especially in light of the results in the convective anvils (high clouds). As at night when no shortwave radiation is reflected and the longwave is the only outgoing radiation, the cirrus anvil extent will likely have a large impact on the TOA outgoing radiation. Was this investigated and see comment below?

**Reply:** The effect of the INP parameterisation on the outgoing radiation in the night-time was not investigated in detail due to the short length of our simulations. When the spin-up period is excluded from the analysis period, less than a full 24 hours of simulation time remains. Thus we don't have data from a full nighttime period. As such, we decided to focus on the daytime hours for which we have simulation data for all hours in question to avoid an arbitrary bias in the results due to simulation length. However, the effect of including the night-time hours (10:00-23:45 UTC) on the outgoing longwave radiation and the cloud fraction was tested and found not to have a large effect on the stated change in the outgoing longwave radiation due to the inclusion of heterogeneous ice nucleation, particularly relative to the very large changes in the outgoing shortwave radiation. This is what we refer to when we say there was little sensitivity to the time frame. The overall outgoing radiation will of course be sensitive to the inclusion of the night-time hours owing to the absence of outgoing shortwave at night-time. This has been clarified in Section 2.1.3.

Change to paper: The following bold text has been inserted in Section 2.1.3. "The sensitivity of the outgoing longwave radiation and the cloud fraction to time period selection was tested and found to have little impact. The overall outgoing radiation (shortwave + longwave) will be sensitive to the time period selection owing to the absence of outgoing shortwave radiation at night-time. We focus on the radiation during daylight hours only because our simulation is only 24 hours in length owing to computational restrictions and therefore when the spin-up period is excluded from the analysis, less than 24 hours of simulation data remains with much of the night-time hours removed with the spin-up period."

As I am unfamiliar with the SOCRATES radiation scheme, do the clouds ever become saturated in the amount of radiation that they can emit? I would think this would occur quiet quickly in the cirrus anvils.

Reply: Clouds do not become saturated in the amount of radiation they can emit. Even for high optical thicknesses, e.g. at the centre of the anvil over the convective core, there should be changes (however small) in the cloud reflectivity due to the cloud properties.

The authors do a nice job of showing that the slope of the INP parameterization used influences the strength of the changes in TOA outgoing radiation. However, as the A13 parameterization has the steepest slope at temperatures above ~-26 C but also no slope at colder temperatures, how does that impact the argument that the slope of the parameterization is critical? Perhaps it is better to state that the number of INPs at cold temperatures is more important than at warmer temperatures in the MPC cloud regime?

Reply: The reviewer is correct that at high dust concentrations the slope of the N12 and A13 parameterisation is flat owing to the plateau in INP number concentrations that occurs once the parameterisations reach the number concentration of dust represented in the model gridbox in question and therefore the number concentrations of aerosols capable of nucleating ice and the parameterisation slope are not decoupled in this work. The relative importance of these two variables is being investigated in current, as yet unpublished, work, and we have added a paragraph detailing this caveat to the limitations section (Section 4).

Change to paper: The following text has been added to Section 4 of the paper: "Furthermore, while many cloud macro- and microphysical were correlated with INP parameterisation slope, the slope of the parameterisation at low temperatures for the A13 and N12 parameterisations can be flat because the parameterisations plateau once they reach the number concentration of dust represented in the model gridbox in question. This means that at high dust concentrations, the slope of the INP parameterisation correlates with the INP concentration at temperatures between -25 and -35°C (Figure 2). This means that the absolute number concentration of aerosols capable of nucleating ice is not decoupled from the INP parameterisation slope in some INP parameterisations and that some cloud responses attributed to changes in the INP parameterisation slope may have in fact been caused by the absolute INP number concentration of aerosols capable of nucleating slope and the absolute number concentration of aerosols capable of nucleating slope and the absolute number concentration of aerosols capable of nucleating ice is not decoupled from the INP parameterisation slope in some INP parameterisations and that some cloud responses attributed to changes in the INP parameterisation slope may have in fact been caused by the absolute INP number concentration of aerosols capable of nucleating ice will be investigated in future work. However, whether the INP number concentration plateaus at cold temperatures is determined in part by the INP parameterisation slope, and correlations with INP parameterisation slope are evident at both warm and cold cloud altitudes indicating the importance of the INP parameterisation slope."

It is mentioned clearly that more studies are needed to investigate what the impacts of adding INPs has over a longer period. However, is it possible to hypothesize on what impacts the increase in TOA outgoing radiation has on subsequent cloud development on subsequent days? More specifically would the reduction in surface temperature reduce the ability of convective clouds to form and ultimately over a long period offset any changes to the TOA outgoing radiation?

Reply: It is certainly possible that reductions in surface temperature due to the radiative changes reported in the paper could affect cloud formation beyond the simulation length. However, due to the complex interactions between multiple hydrometeor and cloud types in these simulations, we don't feel able to give a reliable prediction of the effect of the changes in radiation presented on cloud formation beyond the time period analysed. For example, as noted in our results, while our simulations show an increase in domain cloud fraction within our analysis period due to the inclusion of INP, the anvil cloud fraction is substantially reduced. As anvil cloud can persist in the atmosphere longer than the convective cloud that forms it, it is possible that the reductions in anvil cloud could become more important to the overall cloud signal over a longer time period. Many factors will contribute to changes in the cloud formation in the day(s) after the simulation end including changes in the moisture and temperature profiles, and convergence lines due to large scale flow making speculation difficult. Furthermore, as our domain is over the ocean, the surface temperature will not react very quickly to changes in the radiation.

**Minor comments:**

Line 170: Is the rime mass calculated for the snow hydrometeor class or only for the ice crystal class? Please clarify here as well as in the following comments on this.

**Reply**: The rime mass is calculated from snow and graupel species. We appreciate this was not clear previously and have clarified this wherever necessary.

Change to paper: "The rate of splinter production per rimed mass is prescribed with 350 new ice splinters produced per milligram of rime at -5°C. Splinters are produced from rime mass of snow and graupel."

Line 190: Here it is stated that the radiative cloud properties are not affected by changes in ice or snow number or any changes to rain and graupel. However, based on the following lines it sounds like the cloud radiative properties are sensitive to the mass of these hydrometeor classes. If that is the case, please specify that in this sentence as the mass is sensitive to the number/size and therefore this sentence is potentially misleading. Change to paper: "It does not explicitly consider changes in ice crystal or snow number concentration or size (though changes in number and size will affect mass concentrations which are considered), and does not consider any changes to rain or graupel species."

Line 221: Please clarify why the change in radiation from clear sky areas is only multiplied by the clear sky fraction of the sensitivity run (s) and not the change in the clear sky fraction from the sensitivity run and reference run (s-r), as is done for the cloudy sky fraction.

Reply: The equation in question is actually the combination of two interaction terms. The first  $(\Delta Rad_{cl} \times \Delta cf)$  is the change in radiation caused by changes in cloud albedo that occurs in regions where there was previously no cloud, i.e. the change in domain-mean outgoing radiation that can be attributed to regions of new cloud in simulation s having a different albedo to the cloud that was present in simulation r. The second  $(\Delta Rad_{cs} \times (1 - cf_s))$  is any change in clear sky outgoing radiation (a kind of clear sky albedo), and is applied to all clear sky areas, which accounts for any small bit of cloud mass that may be excluded from our cloudy regions if it falls below the cloud mass threshold. Both of these terms are near zero and negligible to the overall change in outgoing radiation so they were combined into one interaction term. They have now been separated into two different terms now shown in Eq. 3 and Eq. 4.

Change to paper: Section 2.1.3 has been altered with the previous Eq.3 now split into Eq. 3 and Eq.4 to calculate the contribution of the interaction between cloud fraction and cloud albedo changes ( $\Delta Rad_{INT}$ ) and the contribution of clear sky albedo ( $\Delta Rad_{CSKY}$ ) changes separately:

"There is interaction between the outgoing radiation from cloudy regions and cloud fraction changes  $(\Delta Rad_{INT})$  which is calculated in Eq. (3).

$$\Delta Rad_{INT} = \Delta Rad_{cl} \times \Delta cf \tag{3}$$

The contribution of changes in the outgoing radiation from clear sky areas ( $\Delta Rad_{CSKY}$ ) can be calculated as shown in Eq. (4).

$$\Delta Rad_{CSKY} = \Delta Rad_{cs} \times (1 - cf_s) \tag{4}$$

Where  $\Delta Rad_{cs}$  is the change in mean outgoing radiation from clear sky areas between simulations s and r and  $cf_{s}$  is the cloud fraction of simulation s.

The total outgoing radiation difference between simulations s and r ( $\Delta Rad_{s-r}$ ) is therefore as shown in Eq. (5).

$$\Delta Rad_{s-r} = Rad_s - Rad_r = \Delta Rad_{REFL} + \Delta Rad_{CF} + \Delta Rad_{INT} + \Delta Rad_{CSKY}$$
(5)

The interaction term  $\Delta Rad_{INT}$  and the clear sky term ( $\Delta Rad_{CSKY}$ ) were found to be negligible and are therefore ignored for the purposes of this paper."

Line 246: Was cloud base always at temperatures above freezing i.e. was the melting layer always within cloud? If not then omitting rain alone may not be enough to ignore falling precipitation. Please clarify this here.

Reply: Cloud base height distribution shows that cloud bases of low and mixed phase clouds occur predominately between 2.5 and 5 km. Figure 1c in the paper indicates that 0°C occurs at around 5 km. Therefore,

most cloud base heights are at temperatures above freezing, but it is possible as the reviewer notes that some falling precipitation is included in our 'in-cloud' values. We felt it was important to include snow and graupel in the cloud mass calculation for determining where cloud was due to their importance in the deep and dynamic convective clouds represented in the domain. The determination of what qualifies as 'in-cloud 'is uncertain and not well defined in the literature. In order to address this, a number of cloud mass threshold for determining 'in-cloud 'values were tested and these were not found to notably affect the sensitivity of the clouds to the INP parameterisation choice. The inclusion of rain in the in-cloud threshold determination also had a negligible effect. The discussion and results presented in general do not depend strongly on the exact classification of 'in-cloud' values because the radiation calculations refer to the domain-wide calculations of top-of-atmosphere outgoing radiation and precipitating areas will be below cloud and so not likely to affect this. However, we agree that the classification of what qualifies as in-cloud should be explored and its effects examined in future work.

Figure 1. Histograms of cloud top and cloud base height distributions throughout the simulations.

Line 257: please remove the additional "our" before "one of our"

Change to paper: As suggested

Line 274: define DMT ie. Droplet Measurement Technologies

Change to paper: As suggested.

and please provide the specifics on the CDP and CIP e.g. size range of measurements.

Change to paper: The following changes in bold have been made to Section 2.2: *"The aircraft cloud droplet number concentration (CDNC), measured using a Droplet Measurement Technique (DMT) cloud droplet probe (which allows measurement of the cloud droplet size distribution for particles with diameters between 3 and 50 µm (Lloyd et al., 2020)), falls predominantly in the regions of parameter space most highly populated by model data when plotted against vertical wind speed (Fig. A2b)."*

"The observed ICNC was derived from measurements using the DMT Cloud Imaging Probes (CIP-15 and CIP-100, photodetector widths of 15 and 100 μm respectively, both with 64 detector elements)"

Line 281: should SODA2 also include a reference?

Change to paper: We are unaware of a reference for the open source SODA2 code but we have now referenced a recent monograph describing the OAP processing. The following change in bold has been made to Section 2.2: "using the SODA2 (System for OAP (optical array probe) Data Analysis) processing code (McFarquhar et al., 2017)", and the relevant citation has been added to the reference list.

Line 290-291: Please clarify here that all ice-phase hydrometeors contribute to HMP.

Change to paper: "Ice crystals formed via homogeneous freezingand sedimented to lower levels can initiate ice production via the Hallett-Mossop process once converted to snow or graupel"

**Line 311-312: Same here.**

Change to paper: "Bear in mind that SIP was active (SIP\_active) in the simulations summarised in Fig. 4a, including in the NoINP simulation in which the Hallett-Mossop process can be initiated by settling ice-phase hydrometeors (either by settling homogeneously frozen ice crystals subsequently converted to snow or graupel, or by settling snow or graupel formed from homogeneously frozen ice crystals at upper cloud levels), indicating that these cloud systems are sensitive to INP even in the presence of SIP."

Line 334: please add a "to" in "due a reduction"

Change to paper: Sentence has been altered to: "When INP are included in a simulation, snow and cloud droplet water path are enhanced, causing increases in total cloud condensate, despite decreases (in all except A13) in ice crystal water path due to a reduction in ice crystal number and mass concentrations **caused by** a reduction in the availability of cloud droplets for homogeneous freezing."

Line 342-345: Here it is stated that the increase in snow and graupel production due to heterogeneous freezing increases sub-cloud evaporation of rain. However, when looking at Figure A5, the snow mass and graupel mass suggest that the A13 parametrization would lead to the largest amount of available melted mass to increase below cloud humidity. Yet, when looking at Figure A6, A13 has one of the lowest increases in sub-cloud humidity. How is this justified?

Reply: In Figure A4, we can see that A13 has an in-cloud concentration of graupel lower than the other 4 parameterisations. In our CASIM simulations, the density and fallspeed of graupel is higher than that of snow, and therefore graupel is likely to make the largest contributions to falling precipitation. Therefore the lower concentration of graupel in A13 relative to the other INP active simulations may explain why it has the lowest increase in sub-cloud humidity.

Change to paper: The parameters used for particle fallspeed and density are listed in Miltenberger et al. (2018) and this is now referenced in Section 2.1.2: "The parameters used in the representation of the size distribution, density and terminal fall speed velocities of each of the five hydrometeor classes represented by CASIM are shown in Table 2 of Miltenberger et al. (2018)."

Line 352-360: In line with my general comment about the slope of the A13 parameterization, I generally agree that the A13 will produce less ice and therefore remove fewer cloud droplets at low levels in the cloud (below 5

km). However, the number of ice crystal produced at this low-level is quite insignificant to the number of cloud droplets. This can be clearly seen in Fig. 2 where at -6 C (Approx 5 km, assuming dry atmospheric lapse rate) the shallower sloped INP parameterizations (with significantly higher INP concentrations than A13 (~3 orders of magnitude)) still only predict

Figure 2. Change in TOA outgoing longwave (10:00-24:00 UTC) radiation for high (anvil) cloud and the entire simulation domain due to the inclusion of INP (INP-NoINP). A positive value indicates more outgoing radiation when INP are active.

Line 396-399: Again it might be worthwhile to cite previous studies here that have investigated this as well e.g references in Gasparini et al, (2020).

Change to paper: As suggested: "The reduction in anvil extent in the presence of INP is caused by increased liquid consumption at all mixed-phase levels, due to heterogeneous freezing, enhanced SIP and increased graupel and snow production, reducing the availability of cloud droplets for homogeneous freezing (Fig. A4b), reducing ICNC at cloud-top, and reducing cloud anvil extent (Fig. 8g), in agreement with previous studies (e.g. Gasparini et al., 2020; van den Heever et al., 2006; Kärcher and U. Lohmann, 2003; Lohmann and Gasparini, 2017; Phillips et al., 2005, 2007; Storelvmo et al., 2013)."

Line 419: The review paper by Korolev and Leisner, (2020) could be included here.

Change to paper: As suggested.

Line 439-440: What effect does the formation of the "anvil" occurring at warmer temperatures have on the outgoing longwave radiation? This may be important as outgoing radiation is related to the temperature to the 4th power.

Reply: As noted in the manuscript results, the outgoing longwave radiation in C86, M92 and D10 is enhanced by the inclusion of the Hallett-Mossop process and decreased in N12 and A13. The lines in question relate to the change in hydrometeor size in the anvil due to changes in the altitude of complete cloud glaciation (which occurs at lower altitudes for N12 and A13). Therefore the lines in question do not directly relate to the anvil forming at warmer temperatures. We did not look into this issue in detail because the changes in shortwave radiation were so much larger than those of longwave radiation, and so changes in cloud reflectivity and cloud fraction was where we directed our focus. The changes in outgoing longwave radiation due to INP parameterisation slope should be examined in more detail in future work.

Figure 2: It is mentioned in the caption that the INP parametrizations shown are for an aerosol concentration of 8 cm-3. As three of these parameterizations are surface area dependent or at least size dependent (D10 e.g. aerosols > 500 nm) what sized particles are assumed here?

Reply: In Figure 2, the assumed mean particle radius is 0.7  $\mu$ m, which is calculated from the number and mass concentrations of the assumed insoluble dust profile shown in Figure 1. In the model, the surface area of the particles is calculated using the available dust particles and the particle size distribution of CASIM. For the D10 parameterisation, all particles are assumed to be over 0.5  $\mu$ m since dust particles are relatively large and the INP number concentration is calculated using the coarse dust size mode in CASIM.

**Change to paper:** The following sentence has been added to the caption of Figure 2: "*N12 and A13 are calculated assuming a mean dust particle radius of 0.7 \mum. In D10, all particles are assumed to be larger than 0.5 \mum."*

Also, a concentration of 8 cm-3 seems to be an unfair comparison to the observations from the Welti et al, (2018) study as the ambient surface aerosol concentration were likely significantly higher when these INP measurements were conducted. Indeed the modelled and measured surface aerosol are approximately 2 orders of magnitude higher than at the 4 km level. This would put the parameterizations, with the exception of A13, significantly higher than the observations by Welti et al (2018). Or are you relying on the values of the modeled insoluble aerosol concentration to justify the comparison with the Welti et al, (2018) data? This should be expanded upon or at least mentioned. Additionally, if the values are normalized to surface area then this should also be mentioned.

Reply: Comparison is made between the modelled INP and the Welti et al. (2018) INP concentrations because the Welti dataset is from the Cape Verde region where our simulations are based. The modelled insoluble dust profile shown in Figure 1b, which is based on a Met Office Unified Model run with the CLASSIC dust scheme, indicates that much of the aerosol measured at the surface is likely not dust. However we have clarified in the figure legend that the Welti data is measured at the surface and the Price data is measured from aircraft.

Change to paper: The following sentence has been added to the caption of Figure 2: "Note that the Welti et al. (2018). Note that the Welti et al. (2018) dataset is from surface INP measurements at Cape Verde while the Price et al. (2018) dataset is measured from an aircraft flown from Cape Verde."

Furthermore, how was the data from Price et al, (2018) presented? Was it scaled to the surface area of the modelled insoluble aerosol or are these just the absolute concentrations per liter of air reported from the airborne samples?

Reply: Shown is the absolute concentrations per liter of air reported from the airborne samples. However, the calculation of the INP number concentrations was tested using a range of aerosol concentrations (additional to that shown in Figure 2) from the modelled profile and in all cases the parameterisations agree relatively well with the Price et al (2018) data.

Also please fix the Welti et al, (2017) to (2018) in the figure legend.

Change to paper: As suggested.

Lastly, I am not sure it makes sense to include the information that D10 was linearized for the correlation analysis in the figure caption here. Perhaps this is better suited in the text or in the first slope correlation plot.

Change to paper: As suggested, the statement regarding the linearization of D10 has been moved to Section 2.1.4.

**References:**

Gasparini, B., McGraw, Z., Storelvmo, T. and Lohmann, U.: To what extent can cirrus cloud seeding counteract global warming?, Environ. Res. Lett., 15(5), 054002, https://doi.org/10.1088/1748-9326/ab71a3, 2020.

Korolev, A. and Leisner, T.: Review of experimental studies on secondary ice production, Atmospheric Chem. Phys., 1–42, https://doi.org/10.5194/acp-2020-537, 2020.